# A Measure-Theoretic Axiomatisation of Causality

**Junhyung Park**
Empirical Inference Department
MPI for Intelligent Systems
72076 Tübingen, Germany
junhyung.park@tuebingen.mpg.de

**Simon Buchholz**
Empirical Inference Department
MPI for Intelligent Systems
72076 Tübingen, Germany
simon.buchholz@tuebingen.mpg.de

**Bernhard Schölkopf**
Empirical Inference Department
MPI for Intelligent Systems
72076 Tübingen, Germany
bs@tuebingen.mpg.de

**Krikamol Muandet**
CISPA
Helmholtz Center for Information Security
66123 Saarbrücken, Germany
muandet@cispa.de

## Abstract

Causality is a central concept in a wide range of research areas, yet there is still no universally agreed axiomatisation of causality. We view causality both as an extension of probability theory and as a study of *what happens when one intervenes on a system*, and argue in favour of taking Kolmogorov's measure-theoretic axiomatisation of probability as the starting point towards an axiomatisation of causality. To that end, we propose the notion of a *causal space*, consisting of a probability space along with a collection of transition probability kernels, called *causal kernels*, that encode the causal information of the space. Our proposed framework is not only rigorously grounded in measure theory, but it also sheds light on long-standing limitations of existing frameworks including, for example, cycles, latent variables and stochastic processes.

## 1 Introduction

Causal reasoning has been recognised as a hallmark of human and machine intelligence, and in the recent years, the machine learning community has taken up a rapidly growing interest in the subject [46, 53, 54], in particular in representation learning [55, 41, 62, 57, 11, 40] and natural language processing [35, 19]. Causality has also been extensively studied in a wide range of other research domains, including, but not limited to, philosophy [39, 64, 16], psychology [60], statistics [45, 56] including social, biological and medical sciences [51, 31, 29, 26], mechanics and law [6].

The field of causality was born from the observation that probability theory and statistics (Figure 1a) cannot encode the notion of causality, and so we need additional mathematical tools to support the enhanced view of the world involving causality (Figure 1b). Our goal in this paper is to give an axiomatic framework of the forwards direction of Figure 1b, which currently consists of many competing models (see Related Works). As a starting point, we observe that the forwards direction of Figure 1a, i.e. *probability theory*, has a set of axioms based on measure theory (Axioms 2.1) that are widely accepted and used[1], and hence argue that it is natural to take the primitive objects of this framework as the basic building blocks. Despite the fact that all of the existing mathematical frameworks of causality recognise the crucial role that probability plays / should play in any causal

---

[1]Kolmogorov's axiomatisation is without doubt the standard in probability theory. However, we are aware of other, less popular frameworks, for example, one that is more amenable to Bayesian probability [34], one based on game theory [59] and imprecise probabilities [61].

37th Conference on Neural Information Processing Systems (NeurIPS 2023).

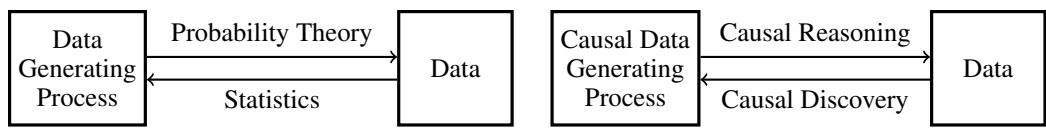

(a) Statistics (or machine learning) is an inverse problem of probability theory.

(b) Causal discovery is an inverse problem of causal reasoning.

Figure 1: Data generating processes and data.

theory, it is surprising that few of them try to build directly upon the axioms of probability theory, and those that do fall short in different ways (see Related Works).

On the other hand, we place *manipulations* at the heart of our approach to causality; in other words, we make changes to some parts of a system, and we are interested in what happens to the rest of this system. This manipulative philosophy towards causality is shared by many philosophers [64], and is the essence behind almost all causal frameworks proposed and adopted in the statistics/machine learning community that we are aware of.

To this end, we propose the notion of *causal spaces* (Definition 2.2), constructed by adding causal objects, called *causal kernels*, directly to probability spaces. We show that causal spaces strictly generalise (the interventional aspects of) existing frameworks, i.e. given any configuration of, for example, a structural causal model or potential outcomes framework, we can construct a causal space that can carry the same (interventional) information. Further, we show that causal spaces can seamlessly support situations where existing frameworks struggle, for example those with hidden confounders, cyclic causal relationships or continuous-time stochastic processes.

**Related Works** We stress that our paper should *not* be understood as a criticism of the existing frameworks, or that somehow our goal is to replace them. On the contrary, we foresee that they will continue to thrive in whatever domains they have been used in, and will continue to find novel application areas.

Most prominently, there are the *structural causal models* (SCMs) [43, 46], based most often on directed acyclic graphs (DAGs). Here, the theory of causality is built around variables and structural equations, and probability only enters the picture through a distribution on the exogeneous variables [33]. Efforts have been made to axiomatise causality based on this framework [21, 23, 28], but models based on structural equations or graphs inevitably rely on assumptions even for the definitions themselves, such as being confined to a finite number of variables, the issue of solvability in the case of non-recursive (or cyclic) cases, that all common causes (whether latent or observed) are modeled, or that the variables in the model do not causally affect anything outside the model. Hence, these cannot be said to be an "axiomatic definition" in the strictest sense. In a series of examples in Section 4, we highlight cases for which causal spaces have a natural representation but SCMs do not, including common causes, cycles and continuous-time stochastic processes.

The *potential outcomes framework* is a competing model, most often used in economics, social sciences or medicine research, in which we have a designated *treatment* variable, whose causal effect we are interested in, and for each value of the treatment variable, we have a separate, *potential outcome* variable [31, 26]. There are other, perhaps lesser-known approaches to model causality, such as that based on decision theory [17, 52], on category theory [32, 20], on an agent explicitly performing actions that transform the state space [15], or settable systems [63].

Perhaps the works that are the most relevant to this paper are those that have already recognised the need for an axiomatisation of causality based on measure-theoretic probability theory. Ortega [42] uses a particular form of a *tree* to define a causal space, and in so doing, uses an alternative, Bayesian set-up of probability theory [34]. It has an obvious drawback that it only considers *countable* sets of "realisations", clearly ruling out many interesting and commonly-occurring cases, and also does not seem to accommodate cycles. Heymann et al. [27] define the *information dependency model* based on measurable spaces to encode causal information. We find this to be a highly interesting and relevant approach, but the issue of cycles and solvability arises, and again, only countable sets of outcomes are considered, with the authors admitting that the results are likely not to hold with uncountable sets. Moreover, probabilities and interventions require additional work to be taken care of. Lastly, Cabreros and Storey [12] attempt to provide a measure-theoretic grounding to the potential

outcomes framework, but thereby confine attention to the setting of a finite number of variables, and even restrict the random variables to be discrete.

Finally, we mention the distinction between *type* causality and *actual* causality. The former is a theory about general causality, involving statements such as "in general, smoking makes lung cancer more likely". Type causality is what we will be concerned with in this paper. Actual causality, on the other hand, is interested in whether a *particular* event was caused by a *particular* action, dealing with statements such as "Bob got lung cancer because he smoked for 30 years". It is an extremely interesting area of research that has far-reaching implications for concepts such as responsibility, blame, law, harm [4, 5], model explanation [7] and algorithmic recourse [36]. Many definitions of actual causality have been proposed [25, 22, 24], but the question of how to define actual causality is still not settled [1]. The current definitions of actual causality are all grounded on (variants) of SCMs, and though out of the scope of this paper, it will be an interesting future research direction to consider how actual causality can be incorporated into our proposed framework.

## 2 Causal Spaces and Interventions

Familiarity with measure-theoretic probability theory is necessary, and we succinctly collect the most essential definitions and results in Appendix A. Most important to note is the definition of a *transition probability kernel* (also given at the end of Appendix A.1): for measurable spaces $(E, \mathcal{E})$ and $(F, \mathcal{F})$, a mapping $K : E \times \mathcal{F} \to [0, \infty]$ is called a *transition probability kernel* from $(E, \mathcal{E})$ into $(F, \mathcal{F})$ if the mapping $x \mapsto K(x, B)$ is measurable for every set $B \in \mathcal{F}$ and the mapping $B \mapsto K(x, B)$ is a probability measure on $(F, \mathcal{F})$ for every $x \in E$. Also worthy of mention is the definition of a *measurable rectangle*: if $(E, \mathcal{E})$ and $(F, \mathcal{F})$ are measurable spaces and $A \in \mathcal{E}$ and $B \in \mathcal{F}$, then the *measurable rectangle* $A \times B$ is the set of all pairs $(x, y)$ with $x \in A$ and $y \in B$. All proofs are deferred to Appendix F. We start by recalling the axioms of probability theory, which we will use as the starting point of our work.

**Axioms 2.1** (Kolmogorov [38]). *A probability space is a triple* $(\Omega, \mathcal{H}, \mathbb{P})$, *where:*

*(i)* $\Omega$ *is a set of outcomes;*

*(ii)* $\mathcal{H}$ *is a collection of events forming a $\sigma$-algebra, i.e. a collection of subsets of $\Omega$ such that (a) $\Omega \in \mathcal{H}$; (b) if $A \in \mathcal{H}$, then $\Omega \setminus A \in \mathcal{H}$; (c) if $A_1, A_2, ... \in \mathcal{H}$, then $\cup_n A_n \in \mathcal{H}$;*

*(iii)* $\mathbb{P}$ *is a probability measure on $(\Omega, \mathcal{H})$, i.e. a function $\mathbb{P} : \mathcal{H} \to [0, 1]$ satisfying (a) $\mathbb{P}(\emptyset) = 0$; (b) $\mathbb{P}(\cup_n A_n) = \sum_n \mathbb{P}(A_n)$ for any disjoint sequence $(A_n)$ in $\mathcal{H}$ (c) $\mathbb{P}(\Omega) = 1$.*

In the development of probability theory, one starts by assuming the existence of a probability space $(\Omega, \mathcal{H}, \mathbb{P})$. However, the actual construction of probability spaces that can carry random variables corresponding to desired random experiments is done through (repeated applications of) two main results – those of Ionescu-Tulcea and Kolmogorov [14, p.160, Chapter IV, Section 4]; the former constructs a probability space that can carry a finite or countably infinite chain of trials, and the latter shows the existence of a probability space that can carry a process with an arbitrary index set. In both cases, the measurable space $(\Omega, \mathcal{H})$ is constructed as a product space:

(i) for a finite set of trials, each taking place in some measurable space $(E_t, \mathcal{E}_t), t = 1, ..., n$, we have $(\Omega, \mathcal{H}) = \otimes_{t=1}^n (E_t, \mathcal{E}_t)$;

(ii) for a countably infinite set of trials, each taking place in some measurable space $(E_t, \mathcal{E}_t)$, $t \in \mathbb{N}$, we have $(\Omega, \mathcal{H}) = \otimes_{t \in \mathbb{N}} (E_t, \mathcal{E}_t)$;

(iii) for a process $\{X_t : t \in T\}$ with an arbitrary index set $T$, we assume that all the $X_t$ live in the same standard measurable space $(E, \mathcal{E})$, and let $(\Omega, \mathcal{H}) = (E, \mathcal{E})^T = \otimes_{t \in T} (E, \mathcal{E})$.

In the construction of a *causal space*, we will take as our starting point a probability space $(\Omega, \mathcal{H}, \mathbb{P})$, where the measure $\mathbb{P}$ is defined on a product measurable space $(\Omega, \mathcal{H}) = \otimes_{t \in T} (E_t, \mathcal{E}_t)$ with the $(E_t, \mathcal{E}_t)$ being the same standard measurable space if $T$ is uncountable. Denote by $\mathcal{P}(T)$ the power set of $T$, and for $S \in \mathcal{P}(T)$, we denote by $\mathcal{H}_S$ the sub-$\sigma$-algebra of $\mathcal{H} = \otimes_{t \in T} \mathcal{E}_t$ generated by measurable rectangles $\times_{t \in T} A_t$, where $A_t \in \mathcal{E}_t$ differs from $E_t$ only for $t \in S$. In particular, $\mathcal{H}_\emptyset = \{\emptyset, \mathcal{H}\}$ is the trivial $\sigma$-algebra of $\Omega = \times_{t \in T} E_t$. Also, we denote by $\Omega_S$ the subspace $\times_{s \in S} E_s$ of $\Omega = \times_{t \in T} E_t$, and for $T \supseteq S \supseteq U$, we let $\pi_{SU}$ denote the natural projection from $\Omega_S$ onto $\Omega_U$.

**Definition 2.2.** A *causal space* is defined as the quadruple $(\Omega, \mathcal{H}, \mathbb{P}, \mathbb{K})$, where $(\Omega, \mathcal{H}, \mathbb{P}) = (\times_{t \in T} E_t, \otimes_{t \in T} \mathcal{E}_t, \mathbb{P})$ is a probability space and $\mathbb{K} = \{K_S : S \in \mathcal{P}(T)\}$, called the *causal mechanism*, is a collection of transition probability kernels $K_S$ from $(\Omega, \mathcal{H}_S)$ into $(\Omega, \mathcal{H})$, called the *causal kernel on $\mathcal{H}_S$*, that satisfy the following axioms:

(i) for all $A \in \mathcal{H}$ and $\omega \in \Omega$,

$$K_\emptyset(\omega, A) = \mathbb{P}(A);$$

(ii) for all $\omega \in \Omega$, $A \in \mathcal{H}_S$ and $B \in \mathcal{H}$,

$$K_S(\omega, A \cap B) = 1_A(\omega) K_S(\omega, B) = \delta_\omega(A) K_S(\omega, B);$$

in particular, for $A \in \mathcal{H}_S$, $K_S(\omega, A) = 1_A(\omega) K_S(\omega, \Omega) = 1_A(\omega) = \delta_\omega(A)$.

Here, the probability measure $\mathbb{P}$ should be viewed as the "observational measure", and the causal mechanism $\mathbb{K}$, consisting of causal kernels $K_S$ for $S \in \mathcal{P}(T)$, contains the "causal information" of the space, by directly specifying the interventional distributions. We write $1_A(\omega)$ when viewed as a function in $\omega$ for a fixed $A$, and $\delta_\omega(A)$ when viewed as a measure for a fixed $\omega \in \Omega$. Note that $\mathbb{K}$ cannot be determined "independently" of the probability measure $\mathbb{P}$, since, for example, $K_\emptyset$ is clearly dependent on $\mathbb{P}$ by (i).

Before we discuss the meaning of the two axioms, we immediately give the definition of an *intervention*. An intervention is carried out on a sub-$\sigma$-algebra of the form $\mathcal{H}_U$ for some $U \in \mathcal{P}(T)$. In the following, for $S \in \mathcal{P}(T)$, we denote $\omega_S = \pi_{TS}(\omega)$. Then note that $\Omega = \Omega_S \times \Omega_{T \setminus S}$ and for any $\omega \in \Omega$, we can decompose it into components as $\omega = (\omega_S, \omega_{T \setminus S})$. Then $K_S(\omega, A) = K_S((\omega_S, \omega_{T \setminus S}), A)$ for any $A \in \mathcal{H}$ only depends on the first $\omega_S$ component of $\omega = (\omega_S, \omega_{T \setminus S})$. As a slight abuse of notation, we will sometimes write $K_S(\omega_S, A)$ for conciseness.

**Definition 2.3.** Let $(\Omega, \mathcal{H}, \mathbb{P}, \mathbb{K}) = (\times_{t \in T} E_t, \otimes_{t \in T} \mathcal{E}_t, \mathbb{P}, \mathbb{K})$ be a causal space, and $U \in \mathcal{P}(T)$, $\mathbb{Q}$ a probability measure on $(\Omega, \mathcal{H}_U)$ and $\mathbb{L} = \{L_V : V \in \mathcal{P}(U)\}$ a causal mechanism on $(\Omega, \mathcal{H}_U, \mathbb{Q})$. An *intervention on $\mathcal{H}_U$ via $(\mathbb{Q}, \mathbb{L})$* is a new causal space $(\Omega, \mathcal{H}, \mathbb{P}^{\mathrm{do}(U, \mathbb{Q})}, \mathbb{K}^{\mathrm{do}(U, \mathbb{Q}, \mathbb{L})})$, where the *intervention measure* $\mathbb{P}^{\mathrm{do}(U, \mathbb{Q})}$ is a probability measure on $(\Omega, \mathcal{H})$ defined, for $A \in \mathcal{H}$, by

$$\mathbb{P}^{\mathrm{do}(U, \mathbb{Q})}(A) = \int \mathbb{Q}(d\omega) K_U(\omega, A) \tag{1}$$

and $\mathbb{K}^{\mathrm{do}(U, \mathbb{Q}, \mathbb{L})} = \{K_S^{\mathrm{do}(U, \mathbb{Q}, \mathbb{L})} : S \in \mathcal{P}(T)\}$ is the *intervention causal mechanism* whose *intervention causal kernels* are

$$K_S^{\mathrm{do}(U, \mathbb{Q}, \mathbb{L})}(\omega, A) = \int L_{S \cap U}(\omega_{S \cap U}, d\omega_U') K_{S \cup U}((\omega_{S \setminus U}, \omega_U'), A). \tag{2}$$

The intuition behind these definitions is as follows. Starting from the probability space $(\Omega, \mathcal{H}, \mathbb{P})$, we choose a "subspace" on which to intervene, namely a sub-$\sigma$-algebra $\mathcal{H}_U$ of $\mathcal{H}$. The *intervention* is the process of placing any desired measure $\mathbb{Q}$ on this "subspace" $(\Omega, \mathcal{H}_U)$, along with an *internal* causal mechanism $\mathbb{L}$ on this "subspace"[2]. The causal kernel $K_U$ corresponding to the "subspace" $\mathcal{H}_U$, which is encoded in the original causal space, determines what the *intervention measure* on the whole space $\mathcal{H}$ will be, via equation (1). For the causal kernels after intervention, the causal effect first takes place within $\mathcal{H}_U$ via the internal causal mechanism $\mathbb{L}$, then propagates to the rest of $\mathcal{H}$ via equation (2).

The definition of intervening on a $\sigma$-algebra of the form $\mathcal{H}_U$ given in Definition 2.3 sheds light on the two axioms of causal spaces given in Definition 2.2.

**Remark 2.4. Trivial Intervention** Axiom (i) of causal mechanisms in Definition 2.2 ensures that intervening on the trivial $\sigma$-algebra (i.e. not intervening at all) leaves the probability measure intact, i.e. writing $\mathbb{Q}$ for the trivial probability measure on $\{\emptyset, \Omega\}$, we have $\mathbb{P}^{\mathrm{do}(\emptyset, \mathbb{Q})} = \mathbb{P}$.

**Interventional Determinism** Axiom (ii) of Definition 2.2 ensures that for any $A \in \mathcal{H}_U$, we have $\mathbb{P}^{\mathrm{do}(U, \mathbb{Q})}(A) = \mathbb{Q}(A)$, which means that if we intervene on the causal space by giving $\mathcal{H}_U$ a particular probability measure $\mathbb{Q}$, then $\mathcal{H}_U$ indeed has that measure with respect to the intervention probability measure.

---

[2]Choosing $\mathbb{Q}$ to have measure 1 on a single element would correspond to what is known as a "hard intervention" in the SCM literature. Letting $\mathbb{Q}$ and $\mathbb{L}$ be arbitrary would allow us to obtain any "soft intervention".

The following example should serve as further clarification of the concepts.

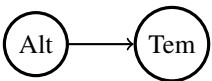

Figure 2: Altitude and Temperature.

**Example 2.5.** *Let $E_1 = E_2 = \mathbb{R}$, and $\mathcal{E}_1, \mathcal{E}_2$ be Lebesgue $\sigma$-algebras on $E_1$ and $E_2$. Each $e_1 \in E_1$ and $e_2 \in E_2$ respectively represent the altitude in metres and temperature in Celsius of a random location. For simplicity, we assume a jointly Gaussian measure $\mathbb{P}$ on $(\Omega, \mathcal{H}) = (E_1 \times E_2, \mathcal{E}_1 \otimes \mathcal{E}_2)$, say with mean vector $\begin{pmatrix} 1000 \\ 10 \end{pmatrix}$ and covariance matrix $\begin{pmatrix} 300 & -15 \\ -15 & 1 \end{pmatrix}$. For each $e_1 \in E_1$ and $A \in \mathcal{E}_2$, we let $K_1(e_1, A)$ be the conditional measure of $\mathbb{P}$ given $e_1$, i.e. Gaussian with mean $\frac{1200 - e_1}{20}$ and variance $\frac{1}{4}$. This represents the fact that, if we intervene by fixing the altitude of a location, then the temperature of that location will be causally affected. However, if we intervene by fixing a temperature of a location, say by manually heating up or cooling down a place, then we expect that this has no causal effect on the altitude of the place. This can be represented by the causal kernel $K_2(e_2, B) = \mathbb{P}(B)$ for each $B \in \mathcal{E}_1$, i.e. Gaussian measure with mean $1000$ and variance $300$, regardless of the value of $e_2$. The corresponding "causal graph" would be Figure 2. If we intervene on $\mathcal{E}_1$ with measure $\delta_{1000}$, i.e. we fix the altitude at $1000m$, then the intervention measure $\mathbb{P}^{\mathrm{do}(1, \delta_{1000})}$ on $(E_2, \mathcal{E}_2)$ would be Gaussian with mean $10$ and variance $\frac{1}{4}$. If we intervene on $\mathcal{E}_2$ with any measure $\mathbb{Q}$, the intervention measure $\mathbb{P}^{\mathrm{do}(2, \mathbb{Q})}$ on $(E_1, \mathcal{E}_1)$ would still be Gaussian with mean $1000$ and variance $300$.*

The following theorem proves that the intervention measure and causal mechanism are indeed valid.

**Theorem 2.6.** *From Definition 2.3, $\mathbb{P}^{\mathrm{do}(U, \mathbb{Q})}$ is indeed a measure on $(\Omega, \mathcal{H})$, and $\mathbb{K}^{\mathrm{do}(U, \mathbb{Q}, \mathbb{L})}$ is indeed a valid causal mechanism on $(\Omega, \mathcal{H}, \mathbb{P}^{\mathrm{do}(U, \mathbb{Q})})$, i.e. they satisfy the axioms of Definition 2.2.*

To end this section, we make a couple of further remarks on the definition of causal spaces.

**Remark 2.7.** (i) We require causal spaces to be built on top of product probability spaces, as opposed to general probability spaces, and causal kernels are defined on sub-$\sigma$-algebras of $\mathcal{H}$ of the form $\mathcal{H}_S$ for $S \in \mathcal{P}(T)$, as opposed to general sub-$\sigma$-algebras of $\mathcal{H}$. This is because, for two events that are not in separate components of a product space, one can always intervene on one of those events in such a way that the measure on the other event will have to change, meaning the causal kernel cannot be decoupled from the intervention itself. For example, in a dice-roll with outcomes $\{1, 2, 3, 4, 5, 6\}$ each with probability $\frac{1}{6}$, if we intervene to give measure $1$ to roll $6$, then the other outcomes are forced to have measure $0$. Only if we consider separate components of product measurable spaces can we set meaningful causal relationships that are decoupled from the act of intervention itself.

(ii) We do not distinguish between interventions that are practically possible and those that are not. For example, the "causal effect of sunlight on the moon's temperature" cannot be measured realistically, as it would require covering up the sun, but the information encoded in the causal kernel would still correspond to what would happen when we cover up the sun.

## 3 Comparison with Existing Frameworks

In this section, we show how causal spaces can encode the interventional aspects of the two most widely-used frameworks of causality, i.e. structural causal models and the potential outcomes.

### 3.1 Structural Causal Models (SCMs)

Consider an SCM in its most basic form, given in the following definition.

**Definition 3.1** ([46, p.83, Definition 6.2]). A structural causal model $\mathfrak{C} = (\mathbf{S}, \tilde{\mathbb{P}})$ consists of a collection $\mathbf{S}$ of $d$ (structural) assignments $X_j := f_j(\mathbf{PA}_j, N_j), j = 1, ..., d$, where $\mathbf{PA}_j \subseteq \{X_1, ..., X_d\} \setminus \{X_j\}$ are called the *parents of* $X_j$ and $N_j$ are the *noise* variables; and a distribution $\tilde{\mathbb{P}}$ over the noise variables such that they are jointly independent.

The graph $\mathcal{G}$ of an SCM is obtained by creating one vertex for each $X_j$ and drawing directed edges from each parent in $\mathbf{PA}_j$ to $X_j$. This graph is assumed to be acyclic.

Below, we show that a unique causal space that corresponds to such an SCM can be constructed.

First, we let the variables $X_j, j = 1, ..., d$ take values in measurable spaces $(E_j, \mathcal{E}_j)$ respectively, and let $(\Omega, \mathcal{H}) = \otimes_{j=1}^d (E_j, \mathcal{E}_j)$. An SCM $\mathfrak{C}$ entails a unique distribution $\mathbb{P}$ over the variables $\mathbf{X} = (X_1, ..., X_d)$ by the propagation of the noise distribution $\tilde{\mathbb{P}}$ through the structural equations $f_j$ [46, p.84, Proposition 6.3], and we take this $\mathbb{P}$ as the observational measure of the causal space. More precisely, assuming $\{1, ..., d\}$ is a topological ordering, we have, for $A_j \in \mathcal{E}_j, j = 1, ..., d$,

$$\mathbb{P}(A_1 \times E_2 \times ... \times E_d) = \tilde{\mathbb{P}}(\{n_1 : f_1(n_1) \in A_1\})$$

$$\mathbb{P}(A_1 \times A_2 \times E_3 \times ... \times E_d) = \tilde{\mathbb{P}}(\{(n_1, n_2) : (f_1(n_1), f_2(f_1(n_1), n_2)) \in A_1 \times A_2\})$$

$$\vdots$$

$$\mathbb{P}(A_1 \times ... \times A_d) = \tilde{\mathbb{P}}(\{(n_1, ..., n_d) : (f_1(n_1), ..., f_d(f_1(n_1), ..., n_d)) \in A_1 \times ... \times A_d\}).$$

Finally, for each $S \in \mathcal{P}(\{1, ..., d\})$ and for each $\omega \in \Omega$, define $f_j^{S,\omega} = f_j$ if $j \notin S$ and $f_j^{S,\omega} = \omega_j$ if $j \in S$. Then we have

$$K_S(\omega, A_1 \times ... \times A_d) = \tilde{\mathbb{P}}(\{(n_1, ..., n_d) : (f_1^{S,\omega}(n_1), ..., f_d^{S,\omega}(f_1^{S,\omega}(n_1), ..., n_d)) \in A_1 \times ... \times A_d\}).$$

This uniquely specifies the causal space $(\Omega, \mathcal{H}, \mathbb{P}, \mathbb{K})$ that corresponds to the SCM $\mathfrak{C}$. While this shows that causal spaces strictly generalise (interventional aspects of) SCMs, there are fundamental philosophical differences between the two approaches, as highlighted in the following remark.

**Remark 3.2.** (i) The "system" in an SCM can be viewed as the collection of all variables $X_1, ..., X_d$, and the "subsystems" the individual variables or the groups of variables. Each *structural equation* $f_j$ encodes how the whole system, when intervened on, affects a subsystem $X_j$, i.e. how the collection of all other variables affects the individual variables (even though, in the end, the equations only depend on the parents). This way of encoding causal effects seems somewhat inconsistent with the philosophy laid out in the Introduction, that we are interested in what happens to the "system" when we intervene on a "subsystem". It also seems inconsistent with the actual action taken, which is to intervene on subsystems, not the whole system, or the parents of a particular variable.

In contrast, the causal kernels encode exactly what happens to the whole system, i.e. what measure we get on the whole measurable space $(\Omega, \mathcal{H})$, when we intervene on a "subsystem", i.e. put a desired measure on a sub-$\sigma$-algebra of $\mathcal{H}^3$.

(ii) The primitive objects of SCMs are the variables $X_j$, the structural equations $f_j$ and the distribution $P_\mathbf{N}$ over the noise variables. The observational distribution, as well as the interventional distributions, are derived from these objects. It turns out that unique existence of observational and interventional distributions are not guaranteed, and can only be shown under the acyclicity assumption or rather stringent and hard-to-verify conditions on the structural equations and the noise distributions [10]. Moreover, it means that the observational and interventional distributions are not decoupled, and rather are linked through the structural equations $f_j$, and as a result, it is not possible to encode the full range of observational and interventional distributions using just the variables of interest (see Example 4.1).

In contrast, in causal spaces, the observational distribution $\mathbb{P}$, as well as the interventional distributions (via the causal kernels), are the primitive objects. Not only does this automatically guarantee their unique existence, but it also allows the interventional distributions (i.e. the causal information) to be completely decoupled from the observational distribution.

(iii) Galles and Pearl [21, Section 3] propose three axioms of counterfactuals based on SCMs (called *causal models* in that paper), namely, composition, effectiveness and reversibility. Even though these three concepts can be carried over to causal spaces, the mathematics through which they are represented needs to be adapted, since the tools that are used in causal spaces are different from those used in causal models of Galles and Pearl [21]. In particular, we work directly with measures as the primitive objects, whereas Galles and Pearl [21] use the structural equations as the primitive objects, and the probabilities only enter through a measure on the exogenous variables. Thus, the three properties can be phrased in the causal space language as follows:

---

[3] In this sense, some philosophy is shared with *generalised structural equation models (GSEMs)* [48].

**Composition** For $S, R \subseteq T$, denote by $\mathbb{Q}'$ the measure on $\mathcal{H}_{S \cup R}$ obtained by restricting $\mathbb{P}^{\mathrm{do}(S, \mathbb{Q})}$. Then $\mathbb{P}^{\mathrm{do}(S, \mathbb{Q})} = \mathbb{P}^{\mathrm{do}(S \cup R, \mathbb{Q}')}$. In words, intervening on $\mathcal{H}_S$ via the measure $\mathbb{Q}$ is the same as intervening on $\mathcal{H}_{S \cup R}$ via the measure that it would have if we intervened on $\mathcal{H}_S$ via $\mathbb{Q}$.

This is not in general true. A counterexample can be demonstrated with a simple SCM, where $X_1$, $X_2$ and $X_3$ causally affect $Y$, in a way that depends not only on the marginal distributions of $X_1$, $X_2$ and $X_3$ but their joint distribution, and $X_1$, $X_2$ and $X_3$ have no causal relationships among them. Then intervening on $X_1$ with some measure $\mathbb{Q}$ cannot be the same as intervening on $X_1$ and $X_2$ with $\mathbb{Q} \otimes \mathbb{P}$, since such an intervention would change the joint distribution of $X_1$, $X_2$ and $X_3$, even if we give them the same marginal distributions.

**Effectiveness** For $S \subseteq R \subseteq T$, if we intervene on $\mathcal{H}_R$ via a measure $\mathbb{Q}$, then $\mathcal{H}_S$ has measure $\mathbb{Q}$ restricted to $\mathcal{H}_S$.

This is indeed guaranteed by interventional determinism (Definition 2.2(ii)), so effectiveness continues to hold in causal spaces.

**Reversibility** For $S, R, U \subseteq T$, let $\mathbb{Q}$ be some measure on $\mathcal{H}_S$, and $\mathbb{Q}_1$ and $\mathbb{Q}_2$ be measures on $\mathcal{H}_{S \cup R}$ and $\mathcal{H}_{S \cup U}$ respectively such that they coincide with $\mathbb{Q}$ when restricted to $\mathcal{H}_S$. Then if $\mathbb{P}^{\mathrm{do}(S \cup R, \mathbb{Q}_1)}(B) = \mathbb{Q}_2(B)$ for all $B \in \mathcal{H}_U$ and if $\mathbb{P}^{\mathrm{do}(S \cup U, \mathbb{Q}_2)}(C) = \mathbb{Q}_1(C)$ for all $C \in \mathcal{H}_R$, then $\mathbb{P}^{\mathrm{do}(S, \mathbb{Q})}(A) = \mathbb{Q}_1(A)$ for all $A \in \mathcal{H}_R$.

This does not hold in general in causal spaces; in fact, Example 4.2 is a counterexample of this, with $S = \emptyset$.

### 3.2 Potential Outcomes (PO) Framework

In the PO framework, the treatment and outcome variables of interest are fixed in advance. Although much of the literature begins with individual units, these units are in the end i.i.d. copies of random variables under the stable unit treatment value assumption (SUTVA), and that is how we begin.

Denote by $(\tilde{\Omega}, \tilde{\mathcal{H}}, \tilde{\mathbb{P}})$ the underlying probability space. Let $Z : \tilde{\Omega} \to \mathcal{Z}$ be the "treatment" variable, taking values in a measurable space $(\mathcal{Z}, \mathfrak{Z})$. Then for each value $z$ of the treatment, there is a separate random variable $Y_z : \tilde{\Omega} \to \mathcal{Y}$, called the "potential outcome given $Z = z$" taking values in a measurable space $(\mathcal{Y}, \mathfrak{Y})$; we also have the "observed outcome", which is the potential outcome consistent with the treatment, i.e. $Y = Y_Z$. The researcher is interested in quantities such as the "average treatment effect", $\tilde{\mathbb{E}}[Y_{z_1} - Y_{z_2}]$, where $\tilde{\mathbb{E}}$ is the expectation with respect to $\tilde{\mathbb{P}}$, to measure the causal effect of the treatment. Often, there are other, "pre-treatment variables" or "covariates", which we denote by $X : \tilde{\Omega} \to \mathcal{X}$, taking values in a measurable space $(\mathcal{X}, \mathfrak{X})$. Given these, another object of interest is the "conditional average treatment effect", defined as $\tilde{\mathbb{E}}[Y_{z_1} - Y_{z_2} \mid X]$.

It is relatively straightforward to construct a causal space that can carry this framework. We define $\Omega = \mathcal{Z} \times \mathcal{Y} \times \mathcal{X}$ and $\mathcal{H} = \mathfrak{Z} \otimes \mathfrak{Y} \otimes \mathfrak{X}$. We also define $\mathbb{P}$, for each $A \in \mathfrak{Z}$, $B \in \mathfrak{Y}$ and $C \in \mathfrak{X}$, as $\mathbb{P}(A \times B \times C) = \tilde{\mathbb{P}}(Z \in A, Y \in B, X \in C)$. As for causal kernels, we are essentially only interested in $K_Z(z, B)$ for $B \in \mathfrak{Y}$, and we define these to be $K_Z(z, B) = \tilde{\mathbb{P}}(Y_z \in B)$.

## 4 Examples

In this section, we give a few more concrete constructions of causal spaces. In particular they are designed to highlight cases which are hard to represent with existing frameworks, but which have natural representations in terms of causal spaces. Comparisons are made particularly with SCMs.

### 4.1 Confounders

The following example highlights the fact that, with graphical models, there is no way to encode correlation but no causation between two variables, using just the variables of interest.

**Example 4.1.** *Consider the popular example of monthly ice cream sales and shark attacks in the US (Figure 3a), that shows that correlation does not imply causation. This cannot be encoded by an SCM with just two variables as in Figure 3b, since no causation means no arrows between the variables, which in turn also means no dependence. One needs to add the common causes into the*

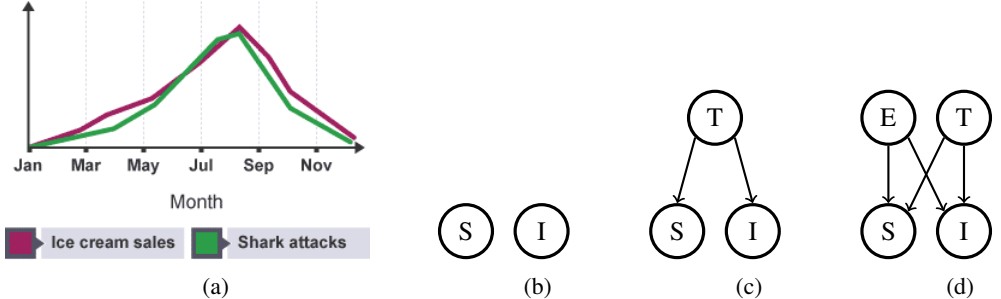

Figure 3: Correlation but no causation between ice-cream sales and shark attacks. S stands for the number of shark attacks, I for ice cream sales, T for temperature and E for economy.

*model (whether observed or latent), the most obvious one being the temperature (high temperatures make people desire ice cream more, as well as to go to the beach more), as seen in Figure 3c. Now we have a model in which both dependence and no causation are captured. But can we stop here? There are probably other factors that affect both variables, such as the economy (the better the economic situation, the more likely people are to buy ice cream, and to take beach holidays) – see Figure 3d. Not only is the model starting to lose parsimony, but as soon as we stop adding variables to the model, we are making an assumption that there are no further confounding variables out there in the world[4].*

*In contrast, causal spaces allow us to model any observational and causal relationships with just the variables that we were interested in, without any restrictions or the need to add more variables. In this particular example, we would take as our causal space $(E_1 \times E_2, \mathcal{E}_1 \otimes \mathcal{E}_2, \mathbb{P}, \mathbb{K})$, where $E_1 = E_2 = \mathbb{R}$ with values in $E_1$ and $E_2$ corresponding to ice cream sales and shark attacks respectively, and $\mathcal{E}_1 = \mathcal{E}_2$ being Lebesgue $\sigma$-algebras. Then we can let $\mathbb{P}$ be a measure that has a strong dependence between any $A \in \mathcal{E}_1$ and $B \in \mathcal{E}_2$, but let the causal kernels be $K_1(x, B) = \mathbb{P}(B)$ for any $x \in E_1$ and $B \in \mathcal{E}_2$, and likewise $K_2(x, A) = \mathbb{P}(A)$ for any $x \in E_2$ and $A \in \mathcal{E}_1$.*

Nancy Cartwright argued against the completeness of causal Markov condition, using an example of two factories [13, p.108], in which there may not even be any confounders between dependent variables, not even an unobserved one. If we accept her position, then there are situations which SCMs would not be able to represent, whereas causal spaces would have no problems at all.

## 4.2 Cycles

As mentioned before, cycles in SCMs cause serious problems, namely that observational and interventional distributions that are consistent with the given structural equations and noise distribution may not exist, and when they do, they may not exist uniquely. This is an artefact of the fact that these distributions are derived from the structural equations rather than taken as the primitive objects. In the vast majority of the cases, cycles are excluded from consideration from the beginning and only directed acyclic graphs (DAGs) are considered. Some works study the *solvability* of cyclic SCMs [23, 10], where the authors investigate under what conditions on the structural equations and the noise variables there exist random variables and distributions that *solve* the given structural equations, and if so, when that happens *uniquely*. Other works have allowed cycles to exist, but restricted the definition of an SCM only to those that have a unique solution [23, 43, 49].

Of course, cyclic causal relationships abound in the real world. In our proposed causal space (Definition 2.2), given two sub-$\sigma$-algebras $\mathcal{H}_S$ and $\mathcal{H}_U$ of $\mathcal{H}$, nothing stops both of them from having a causal effect on the other (see Definition B.1 for a precise definition of causal effects), but we are still guaranteed to have a unique causal space, both before intervention and after intervention on either $\mathcal{H}_S$ or $\mathcal{H}_U$. The following is an example of a situation with "cyclic" causal relationship.

**Example 4.2.** *We want to model the relationship between the amount of rice in the market and its price per kg. Take as the probability space $(E_1 \times E_2, \mathcal{E}_1 \otimes \mathcal{E}_2, \mathbb{P})$, where $E_1 = E_2 = \mathbb{R}$ with values in $E_1$ and $E_2$ representing the amount of rice in the market in million tonnes and the price*

---

[4]One solution could be to add a single "variable" that collects all of the confounders into one, but then the numerical value of this "variable", as well as its distribution and the structural equations from this "variable" into S and I, would be completely meaningless.

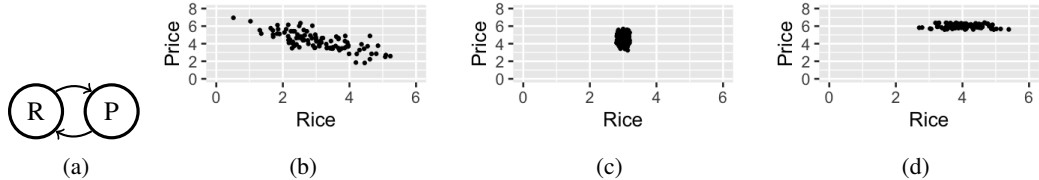

Figure 4: Rice in the market in million tonnes and price per kg in KRW.

*of rice per kg in KRW respectively, $\mathcal{E}_1, \mathcal{E}_2$ are Lebesgue $\sigma$-algebras and $\mathbb{P}$ is for simplicity taken to be jointly Gaussian. Without any intervention, the higher the yield, the more rice there is in the market and lower the price, as in Figure 4b. If the government intervenes on the market by buying up extra rice or releasing rice into the market from its stock, with the goal of stabilising supply at 3 million tonnes, then the price will stabilise accordingly, say with Gaussian distribution with mean 4.5 and standard deviation 0.5, as in Figure 4c. The corresponding causal kernel will be $K_1(3, x) = \frac{2}{\sqrt{2\pi}} e^{-\frac{1}{2} \left( \frac{x - 4.5}{0.5} \right)^2}$. On the other hand, if the government fixes the price of rice at a price, say at 6,000 KRW per kg, then the farmers will be incentivised to produce more, say with Gaussian distribution with mean 4 and standard deviation 0.5, as in Figure 4d. The corresponding causal kernel will be $K_2(6, y) = \frac{2}{\sqrt{2\pi}} e^{-\frac{1}{2} \left( \frac{y - 4}{0.5} \right)^2}$.*

Causal spaces treat causal effects really as what happens after an intervention takes place, and with this approach, cycles can be rather naturally encoded, as shown above. We do not view cyclic causal relationships as an equilibrium of a dynamical system, or require it to be viewed as an acyclic stochastic process, as done by some authors [46, p.85, Remark 6.5].

### 4.3 Continuous-time Stochastic Processes, and Parents

A very well established sub-field of probability theory is the field of stochastic processes, in which the index set representing (most often) time can be either discrete or continuous, and in both cases, infinite. However, most causal models start by assuming a finite number of variables, which immediately rules out considering stochastic processes, and efforts to extend to infinite number of variables usually consider only discrete time steps [46, Chapter 10] or dynamical systems [9, 47, 8, 50]. Since probability spaces have proven to accommodate continuous time stochastic processes in a natural way, it is natural to believe that causal spaces, being built up from probability spaces, should be able to enable the incorporation of the concept of causality in the theory of stochastic processes.

Let $W$ be a totally-ordered set, in particular $W = \mathbb{N} = \{0, 1, ...\}$, $W = \mathbb{Z} = \{..., -2, -1, 0, 1, 2, ...\}$, $W = \mathbb{R}_+ = [0, \infty)$ or $W = \mathbb{R} = (-\infty, \infty)$ considered as the time set. Then, we consider causal spaces of the form $(\Omega, \mathcal{H}, \mathbb{P}, \mathbb{K}) = (\times_{t \in T} E_t, \otimes_{t \in T} \mathcal{E}_t, \mathbb{P}, \mathbb{K})$, where the index set $T$ can be written as $T = W \times \tilde{T}$ for some other index set $\tilde{T}$. The following notion captures the intuition that causation can only go forwards in time.

**Definition 4.3.** Let $(\Omega, \mathcal{H}, \mathbb{P}, \mathbb{K}) = (\times_{t \in T} E_t, \otimes_{t \in T} \mathcal{E}_t, \mathbb{P}, \mathbb{K})$ be a causal space, where the index set $T$ can be written as $T = W \times \tilde{T}$, with $W$ representing time. Then we say that the causal mechanism $\mathbb{K}$ *respects time*, or that $\mathbb{K}$ is a *time-respecting causal mechanism*, if, for all $w_1, w_2 \in W$ with $w_1 < w_2$, we have that $\mathcal{H}_{w_2 \times \tilde{T}}$ has no causal effect (in the sense of Definition B.1) on $\mathcal{H}_{w_1 \times \tilde{T}}$.

In a causal space where the index set $T$ has a time component, the fact that causal mechanism $\mathbb{K}$ respects time means that the past can affect the future, but the future cannot affect the past. This already distinguishes itself from the concept of conditioning – conditioning on the future does have implications for past events. We illustrate this point in the example of a Brownian motion.

**Example 4.4.** *We consider a 1-dimensional Brownian motion. Take $(\times_{t \in \mathbb{R}_+} E_t, \otimes_{t \in \mathbb{R}_+} \mathcal{E}_t, \mathbb{P}, \mathbb{K})$, where, for each $t \in \mathbb{R}_+$, $E_t = \mathbb{R}$ and $\mathcal{E}_t$ is the Lebesgue $\sigma$-algebra, and $\mathbb{P}$ is the Wiener measure. For $s < t$, we have causal kernels $K_s(x, y) = \frac{1}{\sqrt{2\pi(t-s)}} e^{-\frac{1}{2(t-s)}(y-x)^2}$ and $K_t(x, y) = \frac{1}{\sqrt{2\pi s}} e^{-\frac{1}{2s}y^2}$. The former says that, if we intervene by setting the value of the process to $x$ at time $s$, then the process starts again from $x$, whereas the latter says that if we intervene at time $t$, the past values at time $s$ are not affected. On the left-hand plot of Figure 5, we set the value of the process at time 1 to 0. The past values of the process are not affected, and there is a discontinuity at time 1 where the process starts*

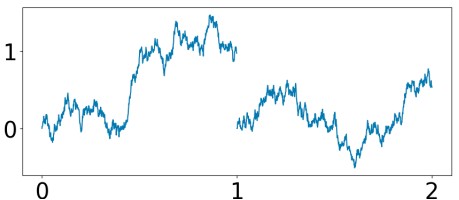 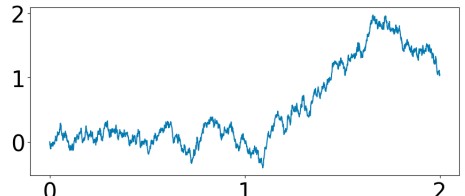

Figure 5: 1-dimensional Brownian motion, intervened and conditioned to have value 0 at time 1.

*again from* 0. *Contrast this to the right-hand plot, where we condition on the process having value* 0 *at time* 1. *This does affect past values, and creates a Brownian bridge from time* 0 *to time* 1.

*Note, Brownian motion is not differentiable, so no approach based on dynamical systems is applicable.*

**Remark 4.5.** The concept of *parents* is central in SCMs – the structural equations are defined on the parents of each variable. However, continuous time is dense, so given two distinct points in time, there is always a time point in between. Suppose we have a one-dimensional continuous time Markov process $(X_t)_{t \in \mathbb{R}}$ [14, p.169], and a point $t_0$ in time. Then for any $t < t_0$, $X_t$ has a causal effect on $X_{t_0}$, but there always exists some $t'$ with $t < t' < t_0$ such that conditioned on $X_{t'}$, $X_t$ does not have a causal effect on $X_{t_0}$, meaning $X_t$ cannot be a parent of $X_{t_0}$. In such a case, $X_{t_0}$ cannot be said to have any parents, and hence no corresponding SCM can be defined.

## 5 Conclusion

In this work, we discussed the lack of a universally agreed axiomatisation of causality, and some arguments as to why measure-theoretic probability theory can provide a good foundation on which to build such an axiomatisation. We proposed *causal spaces*, by enriching probability spaces with causal kernels that encode information about what happens after an intervention. We showed how the interventional aspects of existing frameworks can be captured by causal spaces, and finally we gave some explicit constructions, highlighting cases in which existing frameworks fall short.

Even if causal spaces prove with time to be the correct approach to axiomatise causality, there is much work to be done – in fact, all the more so in that case. Most conspicuously, we only discussed the *interventional* aspects of the theory of causality, but the notion of *counterfactuals* is also seen as a key part of the theory, both *interventional counterfactuals* as advocated by Pearl's ladder of causation [44, Figure 1.2] and *backtracking counterfactuals* [58]. We provide some discussions and possible first steps towards this goal in Appendix E, but mostly leave it as essential future work. Only then will we be able to provide a full comparison with the counterfactual aspects of SCMs and the potential outcomes. As discussed in Section 1, the notion of *actual causality* is also important, and it would be interesting to investigate how this notion can be embedded into causal spaces. Many important definitions, including causal effects, conditional causal effects, hard interventions and sources, as well as technical results, were deferred to the appendix purely due to space constraints, but we foresee that there would be many more interesting results to be proved, inspired both from theory and practice. In particular, the theory of *causal abstraction* [49, 2, 3] should benefit from our proposal, through extensions of homomorphisms of probability spaces to causal spaces[5].

As a final note, we stress again that our goal should *not* be understood as replacing existing frameworks. Indeed, causal spaces cannot compete in terms of interpretability, and in the vast majority of situations in which SCMs, potential outcomes or any of the other existing frameworks are suitable, we expect them to be much more useful. In particular, assumptions are unavoidable for identifiability from observational data, and those assumptions are much better captured by existing frameworks[6], However, just as measure-theoretic probability theory has its value despite not being useful for practitioners in applied statistics, we believe that it is a worthy endeavour to formally axiomatise causality.

---

[5]In a similar vein, Keurti et al. [37] consider homomorphisms between groups of interventions.

[6]Researchers from the potential outcomes community and the graphical model community are arguing as to which framework is better for which situations [30, 43]. We do not take part in this debate.

## Acknowledgments and Disclosure of Funding

We express our sincere gratitude to Robin Evans at the University of Oxford, Michel de Lara at École des Ponts ParisTech and Wojciech Niemiro at the University of Warsaw for fruitful discussions and providing valuable feedback on earlier drafts. We also thank anonymous reviewers for their suggestions for improvements.

This work was supported by the Tübingen AI Center.

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

# A Mathematical Preliminaries

In this section, we recall some basic facts about measure and probability theory that we need for the development in the main body of the paper. We follow Çınlar [14].

## A.1 Measure Theory

Suppose that $E$ is a set. We first define the notion of a $\sigma$-algebra. A non-empty collection $\mathcal{E}$ of $E$ is called a *$\sigma$-algebra* on $E$ if it is closed under complements and countable unions, that is, if

(i) $A \in \mathcal{E} \implies E \backslash A \in \mathcal{E}$;

(ii) $A_1, A_2, ... \in \mathcal{E} \implies \cup_{n=1}^{\infty} A_n \in \mathcal{E}$

[14, p.2]. We call $\{\emptyset, E\}$ the *trivial $\sigma$-algebra* of $E$. If $\mathcal{C}$ is an arbitrary collection of subsets of $E$, then the smallest $\sigma$-algebra that contains $\mathcal{C}$, or equivalently, the intersection of all $\sigma$-algebras that contain $\mathcal{C}$, is called the *$\sigma$-algebra generated by* $\mathcal{C}$, and is denoted $\sigma\mathcal{C}$.

A *measurable space* is a pair $(E, \mathcal{E})$, where $E$ is a set and $\mathcal{E}$ is a $\sigma$-algebra on $E$ [14, p.4].

Suppose $(E, \mathcal{E})$ and $(F, \mathcal{F})$ are measurable spaces. For $A \in \mathcal{E}$ and $B \in \mathcal{F}$, we define the *measurable rectangle $A \times B$* as the set of all pairs $(x, y)$ with $x \in A$ and $y \in B$. We define the *product $\sigma$-algebra* $\mathcal{E} \otimes \mathcal{F}$ on $E \times F$ as the $\sigma$-algebra generated by the collection of all measurable rectangles. The measurable space $(E \times F, \mathcal{E} \otimes \mathcal{F})$ is the *product* of $(E, \mathcal{E})$ and $(F, \mathcal{F})$ [14, p.4]. More generally, if $(E_1, \mathcal{E}_1), ..., (E_n, \mathcal{E}_n)$ are measurable spaces, their product is

$$\bigotimes_{i=1}^{n} (E_i, \mathcal{E}_i) = (\underset{i=1}{\overset{n}{\times}} E_i, \bigotimes_{i=1}^{n} \mathcal{E}_i),$$

where $E_1 \times ... \times E_n$ is the set of all $n$-tuples $(x_1, ..., x_n)$ with $x_i$ in $E_i$ for $i = 1, ..., n$ and $\mathcal{E}_1 \otimes ... \otimes \mathcal{E}_n$ is the $\sigma$-algebra generated by the *measurable rectangles* $A_1 \times ... \times A_n$ with $A_i$ in $\mathcal{E}_i$ for $i = 1, ..., n$ [14, p.44]. If $T$ is an arbitrary (countable or uncountable) index set and $(E_t, \mathcal{E}_t)$ is a measurable space for each $t \in T$, the *product space* of $\{E_t : t \in T\}$ is the set $\times_{t \in T} E_t$ of all collections $(x_t)_{t \in T}$ with $x_t \in E_t$ for each $t \in T$. A rectangle in $\times_{t \in T} E_t$ is a subset of the form

$$\underset{t \in T}{\times} A_t = \{x = (x_t)_{t \in T} \in \underset{t \in T}{\times} E_t : x_t \in A_t \text{ for each } t \text{ in } T\}$$

where $A_t$ differs from $E_t$ for only a finite number of $t$. It is said to be measurable if $A_t \in \mathcal{E}_t$ for every $t$ (for which $A_t$ differs from $E_t$). The $\sigma$-algebra on $\times_{t \in T} E_t$ generated by the collection of all measurable rectangles is called the *product $\sigma$-algebra* and is denoted by $\bigotimes_{t \in T} \mathcal{E}_t$ [14, p.45].

A collection $\mathcal{C}$ of subsets of $E$ is called a p-system if it is closed under intersections [14, p.2]. If two measures $\mu$ and $\nu$ on a measurable space $(E, \mathcal{E})$ with $\mu(E) = \nu(E) < \infty$ agree on a p-system generating $\mathcal{E}$, then $\mu$ and $\nu$ are identical [14, p.16, Proposition 3.7].

Let $(E, \mathcal{E})$ and $(F, \mathcal{F})$ be measurable spaces. A mapping $f : E \to F$ is *measurable* if $f^{-1}B \in \mathcal{E}$ for every $B \in \mathcal{F}$ [14, p.6].

Let $(E, \mathcal{E})$ and $(F, \mathcal{F})$ be measurable spaces. Let $f$ be a bijection between $E$ and $F$, and let $\hat{f}$ denote its functional inverse. Then, $f$ is an *isomorphism* if $f$ is measurable relative to $\mathcal{E}$ and $\mathcal{F}$, and $\hat{f}$ is measurable with respect to $\mathcal{F}$ and $\mathcal{E}$. The measurable spaces $(E, \mathcal{E})$ and $(F, \mathcal{F})$ are *isomorphic* if there exists an isomorphism between them [14, p.11].

A measurable space $(E, \mathcal{E})$ is a *standard measurable space* if it is isomorphic to $(F, \mathcal{B}_F)$ for some Borel subset $F$ of $\mathbb{R}$. Polish spaces with their Borel $\sigma$-algebra are standard measurable spaces [14, p.11].

Let $A \subset E$. Its *indicator*, denoted by $1_A$, is the function defined by

$$1_A(x) = \begin{cases} 1 & \text{if } x \in A \\ 0 & \text{if } x \notin A \end{cases}$$

[14, p.8]. Obviously, $1_A$ is $\mathcal{E}$-measurable if and only if $A \in \mathcal{E}$. A function $f : E \to \mathbb{R}$ is said to be *simple* if it is of the form

$$f = \sum_{i=1}^{n} a_i 1_{A_i}$$

for some $n \in \mathbb{N}$, $a_1, ..., a_n \in \mathbb{R}$ and $A_1, ..., A_n \in \mathcal{E}$ [14, p.8]. The $A_1, ..., A_n \in \mathcal{E}$ can be chosen to be a measurable partition of $E$, and is then called the *canonical form* of the simple function $f$. A positive function on $E$ is $\mathcal{E}$-measurable if and only if it is the limit of an increasing sequence of positive simple functions [14, p.10, Theorem 2.17].

A *measure* on a measurable space $(E, \mathcal{E})$ is a mapping $\mu : \mathcal{E} \to [0, \infty]$ such that

   (i) $\mu(\emptyset) = 0$;
   (ii) $\mu(\cup_{n=1}^{\infty} A_n) = \sum_{n=1}^{\infty} \mu(A_n)$ for every disjoint sequence $(A_n)$ in $\mathcal{E}$

[14, p.14]. A *measure space* is a triplet $(E, \mathcal{E}, \mu)$, where $(E, \mathcal{E})$ is a measurable space and $\mu$ is a measure on it.

A measurable set $B$ is said to be *negligible* if $\mu(B) = 0$, and an arbitrary subset of $E$ is said to be *negligible* if it is contained in a measurable negligible set. The measure space is said to be *complete* if every negligible set is measurable [14, p.17].

Next, we review the notion of integration of a real-valued function $f : E \to \mathbb{R}$ with respect to $\mu$ [14, p.20, Definition 4.3].

   (a) Let $f : E \to [0, \infty]$ be simple. If its canonical form is $f = \sum_{i=1}^{n} a_i 1_{A_i}$ with $a_i \in \mathbb{R}$, then we define

$$\int f d\mu = \sum_{i=1}^{n} a_i \mu(A_i).$$

   (b) Suppose $f : E \to [0, \infty]$ is measurable. Then by above, we have a sequence $(f_n)$ of positive simple functions such that $f_n \to f$ pointwise. Then we define

$$\int f d\mu = \lim_{n \to \infty} \int f_n d\mu,$$

where $\int f_n d\mu$ is defined for each $n$ by (a).

   (c) Suppose $f : E \to [-\infty, \infty]$ is measurable. Then $f^+ = \max\{f, 0\}$ and $f^- = -\min\{f, 0\}$ are measurable and positive, so we can define $\int f^+ d\mu$ and $\int f^- d\mu$ as in (b). Then we define

$$\int f d\mu = \int f^+ d\mu - \int f^- d\mu$$

provided that at least one term on the right be positive. Otherwise, $\int f d\mu$ is undefined. If $\int f^+ d\mu < \infty$ and $\int f^- d\mu < \infty$, then we say that $f$ is *integrable*.

Finally, we review the notion of *transition kernels*, which are crucial in the consideration of conditional distributions. Let $(E, \mathcal{E})$ and $(F, \mathcal{F})$ be measurable spaces. Let $K$ be a mapping $E \times \mathcal{F} \to [0, \infty]$. Then, $K$ is called a *transition kernel* from $(E, \mathcal{E})$ into $(F, \mathcal{F})$ if

   (a) the mapping $x \mapsto K(x, B)$ is measurable for every set $B \in \mathcal{F}$; and
   (b) the mapping $B \mapsto K(x, B)$ is a measure on $(F, \mathcal{F})$ for every $x \in E$.

A transition kernel from $(E, \mathcal{E})$ into $(F, \mathcal{F})$ is called a *probability transition kernel* if $K(x, F) = 1$ for all $x \in E$. A probability transition kernel $K$ from $(E, \mathcal{E})$ into $(E, \mathcal{E})$ is called a *Markov kernel on* $(E, \mathcal{E})$ [14, p.37,39,40].

## A.2  Probability Theory

Now we translate the above measure-theoretic notions into the language of probability theory, and introduce some additional concepts. A *probability space* is a measure space $(\Omega, \mathcal{H}, \mathbb{P})$ such that

$\mathbb{P}(\Omega) = 1$ [14, p.49]. We call $\Omega$ the *sample space*, and each element $\omega \in \Omega$ an *outcome*. We call $\mathcal{H}$ a collection of *events*, and for any $A \in \mathcal{H}$, we read $\mathbb{P}(A)$ as the *probability that the event $A$ occurs* [14, p.50].

A *random variable* taking values in a measurable space $(E, \mathcal{E})$ is a function $X : \Omega \to E$, measurable with respect to $\mathcal{H}$ and $\mathcal{E}$. The *distribution* of $X$ is the measure $\mu$ on $(E, \mathcal{E})$ defined by $\mu(A) = \mathbb{P}(X^{-1}A)$ [14, p.51]. For an arbitrary set $T$, let $X_t$ be a random variable taking values in $(E, \mathcal{E})$ for each $t \in T$. Then the collection $\{X_t : t \in T\}$ is called a *stochastic process* with *state space* $(E, \mathcal{E})$ and *parameter set* $T$ [14, p.53].

Henceforth, random variables are defined on $(\Omega, \mathcal{H}, \mathbb{P})$ and take values in $[-\infty, \infty]$. We define the *expectation* of a random variable $X : \Omega \to [-\infty, \infty]$ as $\mathbb{E}[X] = \int_\Omega X d\mathbb{P}$ [14, p.57-58]. We also define the *conditional expectation* [14, p.140, Definition 1.3]. Suppose $\mathcal{F}$ is a sub-$\sigma$-algebra of $\mathcal{H}$.

(a) Suppose $X$ is a positive random variable. Then the *conditional expectation of $X$ given $\mathcal{F}$* is any positive random variable $\mathbb{E}_\mathcal{F} X$ satisfying

$$\mathbb{E}[VX] = \mathbb{E}\left[V\mathbb{E}_\mathcal{F} X\right]$$

for all $V : \Omega \to [0, \infty]$ measurable with respect to $\mathcal{F}$.

(b) Suppose $X : \Omega \to [-\infty, \infty]$ is a random variable. If $\mathbb{E}[X]$ exists, then we define

$$\mathbb{E}_\mathcal{F} X = \mathbb{E}_\mathcal{F} X^+ - \mathbb{E}_\mathcal{F} X^-,$$

where $\mathbb{E}_\mathcal{F} X^+$ and $\mathbb{E}_\mathcal{F} X^-$ are defined in (a).

Next, we define *conditional probabilities*, and regular versions thereof [14, pp.149-151]. Suppose $H \in \mathcal{H}$, and let $\mathcal{F}$ be a sub-$\sigma$-algebra of $\mathcal{H}$. Then the *conditional probability* of $H$ given $\mathcal{F}$ is defined as

$$\mathbb{P}_\mathcal{F} H = \mathbb{E}_\mathcal{F} 1_H.$$

Let $Q(H)$ be a version of $\mathbb{P}_\mathcal{F} H$ for every $H \in \mathcal{H}$. Then $Q : (\omega, H) \mapsto Q_\omega(H)$ is said to be a *regular version* of the conditional probability $\mathbb{P}_\mathcal{F}$ provided that $Q$ be a probability transition kernel from $(\Omega, \mathcal{F})$ into $(\Omega, \mathcal{H})$. Regular versions exist if $(\Omega, \mathcal{H})$ is a standard measurable space [14, p.151, Theorem 2.7].

The *conditional distribution* of a random variable $X$ given $\mathcal{F}$ is any transition probability kernel $L : (\omega, B) \mapsto L_\omega(B)$ from $(\Omega, \mathcal{F})$ into $(E, \mathcal{E})$ such that

$$P_\mathcal{F}\{Y \in B\} = L(B) \qquad \text{for all } B \in \mathcal{E}.$$

If $(E, \mathcal{E})$ is a standard measurable space, then a version of the conditional distribution of $X$ given $\mathcal{F}$ exists [14, p.151].

Suppose that $T$ is a totally ordered set, i.e. whenever $r, s, t \in T$ with $r < s$ and $s < t$, we have $r < t$ and for any $s, t \in T$, exactly one of $s < t, s = t$ and $t < s$ holds [18, p.62]. For each $t \in T$, let $\mathcal{F}_t$ be a sub-$\sigma$-algebra of $\mathcal{H}$. The family $\mathcal{F} = \{\mathcal{F}_t : t \in T\}$ is called a *filtration* provided that $\mathcal{F}_s \subset \mathcal{F}_t$ for $s < t$ [14, p.79]. A *filtered probability space* $(\Omega, \mathcal{H}, \mathcal{F}, \mathbb{P})$ is a probability space $(\Omega, \mathcal{H}, \mathbb{P})$ endowed with a filtration $\mathcal{F}$.

Finally, we review the notion of *independence* and *conditional independence*. For a fixed integer $n \geq 2$, let $\mathcal{F}_1, ..., \mathcal{F}_n$ be sub-$\sigma$-algebras of $\mathcal{H}$. Then $\{\mathcal{F}_1, ..., \mathcal{F}_n\}$ is called an *independency* if

$$\mathbb{P}(H_1 \cap ... \cap H_n) = \mathbb{P}(H_1)...\mathbb{P}(H_n)$$

for all $H_1 \in \mathcal{F}_1, ..., H_n \in \mathcal{F}_n$. Let $T$ be an arbitrary index set. Let $\mathcal{F}_t$ be a sub-$\sigma$-algebra of $\mathcal{H}$ for each $t \in T$. The collection $\{\mathcal{F}_t : t \in T\}$ is called an *independency* if its every finite subset is an independency [14, p.82].

Moreover, $\mathcal{F}_1, ..., \mathcal{F}_n$ are said to be *conditional independent* given $\mathcal{F}$ if

$$\mathbb{P}_\mathcal{F}(H_1 \cap ... \cap H_n) = \mathbb{P}_\mathcal{F}(H_1)...\mathbb{P}_\mathcal{F}(H_n)$$

for all $H_1 \in \mathcal{F}_1, ..., H_n \in \mathcal{F}_n$ [14, p.158].

# B   Causal Effect

In this section, we define what it means for a sub-$\sigma$-algebra of the form $\mathcal{H}_S$ to have a *causal effect* on an event $A \in \mathcal{H}$.

**Definition B.1.** Let $(\Omega, \mathcal{H}, \mathbb{P}, \mathbb{K}) = (\times_{t \in T} E_t, \otimes_{t \in T} \mathcal{E}_t, \mathbb{P}, \mathbb{K})$ be a causal space, $U \in \mathcal{P}(T)$, $A \in \mathcal{H}$ an event and $\mathcal{F}$ a sub-$\sigma$-algebra of $\mathcal{H}$ (not necessarily of the form $\mathcal{H}_S$ for some $S \in \mathcal{P}(T)$).

(i) If $K_S(\omega, A) = K_{S \setminus U}(\omega, A)$ for all $S \in \mathcal{P}(T)$ and all $\omega \in \Omega$, then we say that $\mathcal{H}_U$ has *no causal effect on $A$*, or that $\mathcal{H}_U$ is *non-causal to $A$*.

   We say that $\mathcal{H}_U$ has *no causal effect on $\mathcal{F}$*, or that $\mathcal{H}_U$ is *non-causal to $\mathcal{F}$*, if, for all $A \in \mathcal{F}$, $\mathcal{H}_U$ has no causal effect on $A$.

(ii) If there exists $\omega \in \Omega$ such that $K_U(\omega, A) \neq \mathbb{P}(A)$, then we say that $\mathcal{H}_U$ has an *active causal effect on $A$*, or that $\mathcal{H}_U$ is *actively causal to $A$*.

   We say that $\mathcal{H}_U$ has an *active causal effect on $\mathcal{F}$*, or that $\mathcal{H}_U$ is *actively causal to $\mathcal{F}$*, if $\mathcal{H}_U$ has an active causal effect on some $A \in \mathcal{F}$.

(iii) Otherwise, we say that $\mathcal{H}_U$ has a *dormant causal effect on $A$*, or that $\mathcal{H}_U$ is *dormantly causal to $A$*.

   We say that $\mathcal{H}_U$ has a *dormant causal effect on $\mathcal{F}$*, or that $\mathcal{H}_U$ is *dormantly causal to $\mathcal{F}$*, if $\mathcal{H}_U$ does not have an active causal effect on any event in $\mathcal{F}$ and there exists $A \in \mathcal{F}$ on which $\mathcal{H}_U$ has a dormant causal effect.

Sometimes, we will say that $\mathcal{H}_U$ has a *causal effect* on $A$ to mean that $\mathcal{H}_U$ has either an active or a dormant causal effect on $A$.

The intuition is as follows. For any $S \in \mathcal{P}(T)$ and any fixed event $A \in \mathcal{H}$, consider the function $\omega_S \mapsto K_S((\omega_{S \cap U}, \omega_{S \setminus U}), A)$. If $\mathcal{H}_U$ has no causal effect on $A$, then it means that the causal kernel does not depend on the $\omega_{S \cap U}$ component of $\omega_S$. Since this has to hold for all $S \in \mathcal{P}(T)$, it means that it is possible to have, for example, $K_U(\omega, A) = \mathbb{P}(A)$ for all $\omega \in \Omega$ and yet for $\mathcal{H}_U$ to have a causal effect on $A$. This would be precisely the case where $\mathcal{H}_U$ has a dormant causal effect on $A$, and it means that, for some $S \in \mathcal{P}(T)$, $\omega_S \mapsto K_S((\omega_{S \cap U}, \omega_{S \setminus U}), A)$ does depend on the $\omega_{S \cap U}$ component.

We collect some straightforward but important special cases in the following remark.

**Remark B.2.**   (a) If $\mathcal{H}_U$ has no causal effect on $A$, then letting $S = U$ in Definition B.1(i) and applying Definition 2.2(i), we can see that, for all $\omega \in \Omega$,

$$K_U(\omega, A) = K_{U \setminus U}(\omega, A) = K_\emptyset(\omega, A) = \mathbb{P}(A).$$

   In particular, this means that $\mathcal{H}_U$ cannot have both no causal effect and active causal effect on $A$.

(b) It is immediate that the trivial $\sigma$-algebra $\mathcal{H}_\emptyset = \{\emptyset, \Omega\}$ has no causal effect on any event $A \in \mathcal{H}$. Conversely, it is also clear that $\mathcal{H}_U$ for any $U \in \mathcal{P}(T)$ has no causal effect on the trivial $\sigma$-algebra.

(c) Let $U \in \mathcal{P}(T)$ and $\mathcal{F}$ a sub-$\sigma$-algebra of $\mathcal{H}$. If $\mathcal{H}_U \cap \mathcal{F} \neq \{\emptyset, \Omega\}$, then $\mathcal{H}_U$ has an active causal effect on $\mathcal{F}$, since, for $A \in \mathcal{H}_U \cap \mathcal{F}$ with $A \neq \emptyset$ and $A \neq \Omega$, Definition 2.2(ii) tells us that $K_U(\cdot, A) = 1_A(\cdot) \neq \mathbb{P}(A)$. In particular, $\mathcal{H}_U$ has an active causal effect on itself. Further, the full $\sigma$-algebra $\mathcal{H} = \mathcal{H}_T$ has an active causal effect on all of its sub-$\sigma$-algebras except the trivial $\sigma$-algebra, and every $\mathcal{H}_U, U \in \mathcal{P}(T)$ except the trivial $\sigma$-algebra has an active causal effect on the full $\sigma$-algebra $\mathcal{H}$.

(d) Let $U \in \mathcal{P}(T)$ and $\mathcal{F}_1, \mathcal{F}_2$ be sub-$\sigma$-algebras of $\mathcal{H}$. If $\mathcal{F}_1 \subseteq \mathcal{F}_2$ and $\mathcal{H}_U$ has no causal effect on $\mathcal{F}_2$, then it is clear that $\mathcal{H}_U$ has no causal effect on $\mathcal{F}_1$.

(e) If $\mathcal{H}_U$ has no causal effect on an event $A$, then for any $V \in \mathcal{P}(T)$ with $V \subseteq U$, $\mathcal{H}_V$ has no causal effect on $A$. Indeed, take any $S \in \mathcal{P}(T)$. Then using the fact that $\mathcal{H}_U$ has no causal effect on $A$, see that, for any $\omega \in \Omega$,

$$K_{S \setminus V}(\omega, A) = K_{(S \setminus V) \setminus U}(\omega, A) \qquad \text{applying Definition B.1(i) with } S \setminus V$$

$$= K_{S\setminus U}(\omega, A) \qquad\qquad \text{since } V \subseteq U$$
$$= K_S(\omega, A) \qquad\qquad \text{applying Definition B.1(i) with } S.$$

Since $S \in \mathcal{P}(T)$ was arbitrary, we have that $\mathcal{H}_V$ has no causal effect on $A$.

(f) Contrapositively, if $U, V \in \mathcal{P}(T)$ with $V \subseteq U$ and $\mathcal{H}_V$ has a causal effect on $A$, then $\mathcal{H}_U$ has a causal effect on $A$.

(g) If $U \in \mathcal{P}(T)$ has no causal effect on $A$, then for any $V \in \mathcal{P}(T)$, we have

$$K_V(\omega, A) = K_{U \cup V}(\omega, A).$$

Indeed,

$$K_{U \cup V}(\omega, A) = K_{(U \cup V)\setminus(U\setminus V)}(\omega, A) \quad \text{since } U \setminus V \text{ has no causal effect on } A \text{ by (e)}$$
$$= K_V(\omega, A) \qquad\qquad \text{since } (U \cup V) \setminus (U \setminus V) = V.$$

(h) If $U, V \in \mathcal{P}(T)$ and neither $\mathcal{H}_U$ nor $\mathcal{H}_V$ has a causal effect on $A$, then $\mathcal{H}_{U \cup V}$ has no causal effect on $A$. Indeed, for any $S \in \mathcal{P}(T)$ and any $\omega \in \Omega$,

$$K_{S\setminus(U \cup V)}(\omega, A) = K_{(S\setminus U)\setminus V}(\omega, A)$$
$$= K_{S\setminus U}(\omega, A) \qquad \text{as } V \text{ has no causal effect on } A$$
$$= K_S(\omega, A) \qquad \text{as } U \text{ has no causal effect on } A.$$

Since $S \in \mathcal{P}(T)$ was arbitrary, $\mathcal{H}_{U \cup V}$ has no causal effect on $A$.

(i) Contrapositively, if $U, V \in \mathcal{P}(T)$ and $\mathcal{H}_{U \cup V}$ has a causal effect on $A$, then either $\mathcal{H}_U$ or $\mathcal{H}_V$ has a causal effect on $A$.

Following the definition of no causal effect, we define the notion of a *trivial causal kernel*.

**Definition B.3.** Let $(\Omega, \mathcal{H}, \mathbb{P}, \mathbb{K}) = (\times_{t \in T} E_t, \otimes_{t \in T} \mathcal{E}_t, \mathbb{P}, \mathbb{K})$ be a causal space, and $U \in \mathcal{P}(T)$. We say that the causal kernel $K_U$ is *trivial* if $\mathcal{H}_U$ has no causal effect on $\mathcal{H}_{T\setminus U}$.

Note that we can decompose $\mathcal{H}$ as $\mathcal{H} = \mathcal{H}_U \otimes \mathcal{H}_{T\setminus U}$, and so $\mathcal{H}$ is generated by events of the form $A \times B$ for $A \in \mathcal{H}_U$ and $B \in \mathcal{H}_{T\setminus U}$. But if $K_U$ is trivial, then we have, by Axiom 2.2(ii), $K_U(\omega, A \times B) = 1_A(\omega)\mathbb{P}(B)$ for such a rectangle.

We also define a "conditional" version of causal effects.

**Definition B.4.** Let $(\Omega, \mathcal{H}, \mathbb{P}, \mathbb{K}) = (\times_{t \in T} E_t, \otimes_{t \in T} \mathcal{E}_t, \mathbb{P}, \mathbb{K})$ be a causal space, $U, V \in \mathcal{P}(T)$, $A \in \mathcal{H}$ an event and $\mathcal{F}$ a sub-$\sigma$-algebra of $\mathcal{H}$ (not necessarily of the form $\mathcal{H}_S$ for some $S \in \mathcal{P}(T)$).

(i) If $K_{S \cup V}(\omega, A) = K_{(S \cup V)\setminus(U\setminus V)}(\omega, A)$ for all $S \in \mathcal{P}(T)$ and all $\omega \in \Omega$, then we say that $\mathcal{H}_U$ has *no causal effect on $A$ given $\mathcal{H}_V$*, or that $\mathcal{H}_U$ is *non-causal to $A$ given $\mathcal{H}_V$*.

We say that $\mathcal{H}_U$ has *no causal effect on $\mathcal{F}$ given $\mathcal{H}_V$*, or that $\mathcal{H}_U$ is *non-causal to $\mathcal{F}$ given $\mathcal{H}_V$*, if, for all $A \in \mathcal{F}$, $\mathcal{H}_U$ has no causal effect on $A$ given $\mathcal{H}_V$.

(ii) If there exists $\omega \in \Omega$ such that $K_{U \cup V}(\omega, A) \neq K_V(\omega, A)$, then we say that $\mathcal{H}_U$ has an *active causal effect on $A$ given $\mathcal{H}_V$*, or that $\mathcal{H}_U$ is *actively causal to $A$ given $\mathcal{H}_V$*.

We say that $\mathcal{H}_U$ has an *active causal effect on $\mathcal{F}$ given $\mathcal{H}_V$*, or that $\mathcal{H}_U$ is *actively causal to $\mathcal{F}$ given $\mathcal{H}_V$*, if $\mathcal{H}_U$ has an active causal effect on some $A \in \mathcal{F}$.

(iii) Otherwise, we say that $\mathcal{H}_U$ has a *dormant causal effect on $A$ given $\mathcal{H}_V$*, or that $\mathcal{H}_U$ is *dormantly causal to $A$ given $\mathcal{H}_V$*.

We say that $\mathcal{H}_U$ has a *dormant causal effect on $\mathcal{F}$ given $\mathcal{H}_V$*, or that $\mathcal{H}_U$ is *dormantly causal to $\mathcal{F}$ given $\mathcal{H}_V$*, if $\mathcal{H}_U$ does not have an active causal effect on any event in $\mathcal{F}$ given $\mathcal{H}_V$ and there exists $A \in \mathcal{F}$ on which $\mathcal{H}_U$ has a dormant causal effect given $\mathcal{H}_V$.

Sometimes, we will say that $\mathcal{H}_U$ has a *causal effect on $A$ given $\mathcal{H}_V$* to mean that $\mathcal{H}_U$ has either an active or a dormant causal effect on $A$ given $\mathcal{H}_V$.

The intuition is as follows. For any fixed $S \in \mathcal{P}(T)$ and any fixed event $A \in \mathcal{H}$. consider the function $\omega_{S \cup V} \mapsto K_{S \cup V}((\omega_{(S \cup V) \setminus (U \setminus V)}, \omega_{S \cap (U \setminus V)}), A)$. If $\mathcal{H}_U$ has no causal effect on $A$ given $\mathcal{H}_V$, then it means that the causal kernel does not depend on the $\omega_{S \cap (U \setminus V)}$ component of $\omega_{S \cup V}$; in other words, $\mathcal{H}_U$ only has an influence on $A$ through its $V$ component.

We collect some important special cases in the following remark.

**Remark B.5.**     (a) Letting $V = U$, we always have $K_{S \cup U}(\omega, A) = K_{(S \cup U) \setminus (U \setminus U)}(\omega, A) = K_{S \cup U}(\omega, A)$ for all $\omega \in \Omega$ and $A \in \mathcal{H}$, which means that $\mathcal{H}_U$ has no causal effect on any event $A \in \mathcal{H}$ given itself.

(b) If $\mathcal{H}_U$ has no causal effect on $A$ given $\mathcal{H}_V$, then letting $U = S$ in Definition B.4(i), we see that, for all $\omega \in \Omega$,
$$K_{U \cup V}(\omega, A) = K_V(\omega, A).$$
In particular, this means that $\mathcal{H}_U$ cannot have both no causal effect and active causal effect on $A$ given $\mathcal{H}_V$.

(c) The case $V = \emptyset$ reduces Definition B.4 to Definition B.1, i.e. $\mathcal{H}_U$ having no causal effect in the sense of Definition B.1 is the same as $\mathcal{H}_U$ having no causal effect given $\{\emptyset, \Omega\}$ in the sense of Definition B.4, etc.

(d) It is possible for $\mathcal{H}_U$ to be causal to an event $A$, and for there to exist $V \in \mathcal{P}(T)$ such that $\mathcal{H}_U$ has no causal effect on $A$ given $\mathcal{H}_V$. However, if $\mathcal{H}_U$ has no causal effect on $A$, then for any $V \in \mathcal{P}(T)$, $\mathcal{H}_U$ has no causal effect on $A$ given $\mathcal{H}_V$. To see this, note that Remark B.2(e) tells us that $U \setminus V$ also does not have any causal effect on $A$. Then given any $S \in \mathcal{P}(T)$,
$$K_{S \cup V}(\omega, A) = K_{(S \cup V) \setminus (U \setminus V)}(\omega, A),$$
applying Definition B.1(i) to $S \cup V$. Since $S \in \mathcal{P}(T)$ was arbitrary, $\mathcal{H}_U$ has no causal effect on $A$ given $\mathcal{H}_V$.

## C   Interventions

In this section, we provide a few more definitions and results related to the notion of interventions, introduced in Definition 2.3.

First, we make a few remarks on how the intervention causal kernels $K_S^{\mathrm{do}(U, \mathbb{Q}, \mathbb{L})}$ behave in some special cases, depending on the relationship between $U$ and $S$.

**Remark C.1.**     (a) For $S \in \mathcal{P}(T)$ with $U \subseteq S$, we have, for all $\omega \in \Omega$ and all $A \in \mathcal{H}$,

$$K_S^{\mathrm{do}(U, \mathbb{Q}, \mathbb{L})}(\omega, A) = \int L_U(\omega_U, d\omega_U') K_S((\omega_{S \setminus U}, \omega_U'), A)$$
$$= \int \delta_{\omega_U}(d\omega_U') K_S((\omega_{S \setminus U}, \omega_U'), A) \quad \text{by Definition 2.2(ii)}$$
$$= K_S((\omega_{S \setminus U}, \omega_U), A)$$
$$= K_S(\omega, A).$$

This means that, after an intervention on $\mathcal{H}_U$, subsequent interventions on $\mathcal{H}_S$ with $\mathcal{H}_U \subseteq \mathcal{H}_S$ simply overwrite the original intervention. Note that this is reminiscent of the "partial ordering on the set of interventions" in [49], but in our setting, this is given by the partial ordering induced by the inclusion structure of sub-$\sigma$-algebras of $\mathcal{H}$.

(b) For $S \in \mathcal{P}(T)$ with $S \subseteq U$,

$$K_S^{\mathrm{do}(U, \mathbb{Q}, \mathbb{L})}(\omega, A) = \int L_S(\omega_S, d\omega_U') K_U(\omega_U', A)$$

for all $\omega \in \Omega$ and $A \in \mathcal{H}$, i.e. $K_S^{\mathrm{do}(U, \mathbb{Q}, \mathbb{L})}$ is a product of the two kernels $K_U$ and $L_S$ [14, p.39]; in particular, $K_S^{\mathrm{do}(U, \mathbb{Q}, \mathbb{L})}(\omega, A) = L_S(\omega, A)$ for all $A \in \mathcal{H}_U$.

(c) For $S \in \mathcal{P}(T)$ with $S \cap U = \emptyset$,

$$K_S^{\text{do}(U,\mathbb{Q},\mathbb{L})}(\omega, A) = \int L_\emptyset(\omega_\emptyset, d\omega'_U) K_{S \cup U}((\omega_S, \omega'_U), A)$$

$$= \int \mathbb{Q}(d\omega'_U) K_{S \cup U}((\omega_S, \omega'_U), A) \qquad \text{by Definition 2.2(i)}$$

for all $\omega \in \Omega$ and $A \in \mathcal{H}$, i.e. the effect of intervening on $\mathcal{H}_U$ with $\mathbb{Q}$ then $\mathcal{H}_S$ is the same as intervening on $\mathcal{H}_{U \cup S}$ with a product measure of $\mathbb{Q}$ on $\mathcal{H}_U$ and whatever measure we place on $\mathcal{H}_S$.

We give it a name for the special case in which the internal causal kernels are all trivial (see Definition B.3).

**Definition C.2.** Let $(\Omega, \mathcal{H}, \mathbb{P}, \mathbb{K}) = (\times_{t \in T} E_t, \otimes_{t \in T} \mathcal{E}_t, \mathbb{P}, \mathbb{K})$ be a causal space, and $U \in \mathcal{P}(T)$ and $\mathbb{Q}$ a probability measure on $(\Omega, \mathcal{H}_U)$. A *hard intervention on $\mathcal{H}_U$ via $\mathbb{Q}$* is a new causal space $(\Omega, \mathcal{H}, \mathbb{P}^{\text{do}(U,\mathbb{Q})}, \mathbb{K}^{\text{do}(U,\mathbb{Q},\text{hard})})$, where the intervention measure $\mathbb{P}^{\text{do}(U,\mathbb{Q})}$ is a probability measure $(\Omega, \mathcal{H})$ defined in the same way as in Definition 2.3, and the intervention causal mechanism $\mathbb{K}^{\text{do}(U,\mathbb{Q},\text{hard})} = \{K_S^{\text{do}(U,\mathbb{Q},\text{hard})} : S \in \mathcal{P}(T)\}$ consists of causal kernels that are obtained from the intervention causal kernels in Definition 2.3 in which $L_{S \cap U}$ is a trivial causal kernel, i.e. one that has no causal effect on $\mathcal{H}_{U \setminus S}$.

From the discussion following Definition B.3, we have that, for $A \in \mathcal{H}_{S \cap U}$ and $B \in \mathcal{H}_{U \setminus S}$, $L_{S \cap U}(\omega, A \times B) = 1_A(\omega_{S \cap U})\mathbb{Q}(B)$.

The next result gives an explicit expression for the causal kernels obtained after a hard intervention.

**Theorem C.3.** *Let $(\Omega, \mathcal{H}, \mathbb{P}, \mathbb{K}) = (\times_{t \in T} E_t, \otimes_{t \in T} \mathcal{E}_t, \mathbb{P}, \mathbb{K})$ be a causal space, and $U \in \mathcal{P}(T)$ and $\mathbb{Q}$ a probability measure on $(\Omega, \mathcal{H}_U)$. Then after a hard intervention on $\mathcal{H}_U$ via $\mathbb{Q}$, the intervention causal kernels $K_S^{\text{do}(U,\mathbb{Q},\text{hard})}$ are given by*

$$K_S^{\text{do}(U,\mathbb{Q},\text{hard})}(\omega, A) = K_S^{\text{do}(U,\mathbb{Q},\text{hard})}(\omega_S, A) = \int \mathbb{Q}(d\omega'_{U \setminus S}) K_{S \cup U}((\omega_S, \omega'_{U \setminus S}), A).$$

Intuitively, hard interventions do not encode any internal causal relationships within $\mathcal{H}_U$, so after we subsequently intervene on $\mathcal{H}_S$, the measure $\mathbb{Q}$ that we originally imposed on $\mathcal{H}_U$ remains on $\mathcal{H}_{U \setminus S}$.

The following lemma contains a couple of results about particular sub-$\sigma$-algebras having no causal effects on particular events in the intervention causal space, regardless of the measure and causal mechanism that was used for the intervention.

**Lemma C.4.** *Let $(\Omega, \mathcal{H}, \mathbb{P}, \mathbb{K}) = (\times_{t \in T} E_t, \otimes_{t \in T} \mathcal{E}_t, \mathbb{P}, \mathbb{K})$ be a causal space, and $U \in \mathcal{P}(T)$, $\mathbb{Q}$ a probability measure on $(\Omega, \mathcal{H}_U)$ and $\mathbb{L} = \{L_V : V \in \mathcal{P}(U)\}$ a causal mechanism on $(\Omega, \mathcal{H}_U, \mathbb{Q})$. Suppose we intervene on $\mathcal{H}_U$ via $(\mathbb{Q}, \mathbb{L})$.*

   *(i) For $A \in \mathcal{H}_U$ and $V \in \mathcal{P}(T)$ with $V \cap U = \emptyset$, $\mathcal{H}_V$ has no causal effect on $A$ (c.f. Definition B.1(i)) in the intervention causal space $(\Omega, \mathcal{H}, \mathbb{P}^{\text{do}(U,\mathbb{Q})}, \mathbb{K}^{\text{do}(U,\mathbb{Q},\mathbb{L})})$, i.e. events in the $\sigma$-algebra $\mathcal{H}_U$ on which intervention took place are not causally affected by $\sigma$-algebras outside $\mathcal{H}_U$.*

   *(ii) Again, let $V \in \mathcal{P}(T)$ with $V \cap U = \emptyset$, and also let $A \in \mathcal{H}$ be any event. If, in the original causal space, $\mathcal{H}_V$ had no causal effect on $A$, then in the intervention causal space, $\mathcal{H}_V$ has no causal effect on $A$ either.*

   *(iii) Now let $V \in \mathcal{P}(T)$, $A \in \mathcal{H}$ any event and suppose that the intervention on $\mathcal{H}_U$ via $\mathbb{Q}$ is hard. Then if $\mathcal{H}_V$ had no causal effect on $A$ in the original causal space, then $\mathcal{H}_V$ has no causal effect on $A$ in the intervention causal space either.*

Lemma C.4(ii) and (iii) tell us that, if $\mathcal{H}_V$ had no causal effect on $A$ in the original causal space, then by intervening on $\mathcal{H}_U$ with $V \cap U = \emptyset$ or by any hard intervention, we cannot create a causal effect from $\mathcal{H}_v$ on $A$. However, by intervening on a sub-$\sigma$-algebra that contains both $\mathcal{H}_V$ and (a part of) $A$, and manipulating the internal causal mechanism $\mathbb{L}$ appropriately, it is clear that we can create a causal effect from $\mathcal{H}_V$.

The next result tells us that if a sub-$\sigma$-algebra $\mathcal{H}_U$ has a dormant causal effect on an event $A$, then there is a sub-$\sigma$-algebra of $\mathcal{H}_U$ and a hard intervention after which that sub-$\sigma$-algebra has an active causal effect on $A$.

**Lemma C.5.** *Let $(\Omega, \mathcal{H}, \mathbb{P}, \mathbb{K}) = (\times_{t \in T} E_t, \otimes_{t \in T} \mathcal{E}_t, \mathbb{P}, \mathbb{K})$ be a causal space, and $U \in \mathcal{P}(T)$. For an event $A \in \mathcal{H}$, if $\mathcal{H}_U$ has a dormant causal effect on $A$ in the original causal space, then there exists a hard intervention and a subset $V \subseteq U$ such that in the intervention causal space, $\mathcal{H}_V$ has an active causal effect on $A$.*

The next result is about what happens to a causal effect of a sub-$\sigma$-algebra that has no causal effect on an event conditioned on another sub-$\sigma$-algebra, after intervening on that sub-$\sigma$-algebra.

**Lemma C.6.** *Let $(\Omega, \mathcal{H}, \mathbb{P}, \mathbb{K}) = (\times_{t \in T} E_t, \otimes_{t \in T} \mathcal{E}_t, \mathbb{P}, \mathbb{K})$ be a causal space, and $U, V \in \mathcal{P}(T)$. For an event $A \in \mathcal{H}$, suppose that $\mathcal{H}_U$ has no causal effect on $A$ given $\mathcal{H}_V$ (see Definition B.4). Then after an intervention on $\mathcal{H}_V$ via any $(\mathbb{Q}, \mathbb{L})$, $\mathcal{H}_{U \setminus V}$ has no causal effect on $A$.*

The next result shows that, under a hard intervention, a time-respecting causal mechanism stays time-respecting.

**Theorem C.7.** *Let $(\Omega, \mathcal{H}, \mathbb{P}, \mathbb{K}) = (\times_{t \in T} E_t, \otimes_{t \in T} \mathcal{E}_t, \mathbb{P}, \mathbb{K})$ be a causal space, where the index set $T$ can be written as $T = W \times \tilde{T}$, with $W$ representing time and $\mathbb{K}$ respecting time. Take any $U \in \mathcal{P}(T)$ and any probability measure $\mathbb{Q}$ on $\mathcal{H}_U$. Then the intervention causal mechanism $\mathbb{K}^{\mathrm{do}(U, \mathbb{Q}, \mathrm{hard})}$ also respects time.*

## D   Sources

In causal spaces, the observational distribution $\mathbb{P}$ and the causal mechanism $\mathbb{K}$ are completely decoupled. In Section 3.1, we give a detailed argument as to why this is desirable, but of course, there is no doubt that the special case in which the causal kernels coincide with conditional measures with respect to $\mathbb{P}$ is worth studying. To that end, we introduce the notion of *sources*.

**Definition D.1.** Let $(\Omega, \mathcal{H}, \mathbb{P}, \mathbb{K}) = (\times_{t \in T} E_t, \otimes_{t \in T} \mathcal{E}_t, \mathbb{P}, \mathbb{K})$ be a causal space, $U \in \mathcal{P}(T)$, $A \in \mathcal{H}$ an event and $\mathcal{F}$ a sub-$\sigma$-algebra of $\mathcal{H}$. We say that $\mathcal{H}_U$ is a *(local) source* of $A$ if $K_U(\cdot, A)$ is a version of the conditional probability $\mathbb{P}_{\mathcal{H}_U}(A)$. We say that $\mathcal{H}_U$ is a *(local) source* of $\mathcal{F}$ if $\mathcal{H}_U$ is a source of all $A \in \mathcal{F}$. We say that $\mathcal{H}_U$ is a *global source* of the causal space if $\mathcal{H}_U$ is a source of all $A \in \mathcal{H}$.

Clearly, source $\sigma$-algebras are not unique (whether local or global). It is easy to see that $\mathcal{H}_\emptyset = \{\emptyset, \Omega\}$ and $\mathcal{H} = \mathcal{H}_T = \otimes_{t \in T} \mathcal{E}_t$ are global sources, and axiom (ii) of Definition 2.2 implies that any $\mathcal{H}_S$ is a local source of any of its sub-$\sigma$-algebras, including itself, since, for any $A \in \mathcal{H}_U$, $\mathbb{P}_{\mathcal{H}_U}(A) = 1_A$. Also, a sub-$\sigma$-algebra of a source is not necessarily a source, nor is a $\sigma$-algebra that contains a source necessarily a source (whether local or global). In Example 2.5 above, altitude is a source of temperature (and hence a global source), since the causal kernel corresponding to temperature coincides with the conditional measure given altitude, but temperature is not a source of altitude.

When we intervene on $\mathcal{H}_U$ (via any $(\mathbb{Q}, \mathbb{L})$), $\mathcal{H}_U$ becomes a global source. This precisely coincides with the "gold standard" that is randomised control trials in causal inference, i.e. the idea that, if we are able to intervene on $\mathcal{H}_U$, then the causal effect of $\mathcal{H}_U$ on any event can be obtained by first intervening on $\mathcal{H}_U$, then considering the conditional distribution on $\mathcal{H}_U$. Next is a theorem showing that when one intervenes on $\mathcal{H}_U$, then $\mathcal{H}_U$ becomes a source.

**Theorem D.2.** *Suppose we have a causal space $(\Omega, \mathcal{H}, \mathbb{P}, \mathbb{K}) = (\times_{t \in T} E_t, \otimes_{t \in T} \mathcal{E}_t, \mathbb{P}, \mathbb{K})$, and let $U \in \mathcal{P}(T)$.*

   *(i) For any measure $\mathbb{Q}$ on $\mathcal{H}_U$ and any causal mechanism $\mathbb{L}$ on $(\Omega, \mathcal{H}_U, \mathbb{Q})$, the causal kernel $K_U^{\mathrm{do}(U, \mathbb{Q}, \mathbb{L})} = K_U$ is a version of $\mathbb{P}_{\mathcal{H}_U}^{\mathrm{do}(U, \mathbb{Q})}$, which means that $\mathcal{H}_U$ is a global source $\sigma$-algebra of the intervened causal space $(\Omega, \mathcal{H}, \mathbb{P}^{\mathrm{do}(U, \mathbb{Q})}, \mathbb{K}^{\mathrm{do}(U, \mathbb{Q}, \mathbb{L})})$.*

   *(ii) Suppose $V \in \mathcal{P}(T)$ with $V \subseteq U$. Suppose that the measure $\mathbb{Q}$ on $(\Omega, \mathcal{H}_U)$ factorises over $\mathcal{H}_V$ and $\mathcal{H}_{U \setminus V}$, i.e. for any $A \in \mathcal{H}_V$ and $B \in \mathcal{H}_{U \setminus V}$, $\mathbb{Q}(A \cap B) = \mathbb{Q}(A)\mathbb{Q}(B)$. Then after a hard intervention on $\mathcal{H}_U$ via $\mathbb{Q}$, the causal kernel $K_V^{\mathrm{do}(U, \mathbb{Q}, \mathrm{hard})}$ is a version of $\mathbb{P}_V^{\mathrm{do}(U, \mathbb{Q})}$, which means that $\mathcal{H}_V$ is a global source $\sigma$-algebra of the intervened causal space $(\Omega, \mathcal{H}, \mathbb{P}^{\mathrm{do}(U, \mathbb{Q})}, \mathbb{K}^{\mathrm{do}(U, \mathbb{Q}, \mathrm{hard})})$.*

Let $A \in \mathcal{H}$ be an event, and $U \in \mathcal{P}(T)$. By the definition of the intervention measure (Definition 2.3), we always have

$$\mathbb{P}^{\mathrm{do}(U,\mathbb{Q})}(A) = \int \mathbb{Q}(d\omega) K_U(\omega, A),$$

hence $\mathbb{P}^{\mathrm{do}(U,\mathbb{Q})}(A)$ can be written in terms of $\mathbb{P}$ and $\mathbb{Q}$ if $K_U(\omega, A)$ can be written in terms of $\mathbb{P}$. This can be seen to occur in three trivial cases: first, if $\mathcal{H}_U$ is a local source of $A$ (see Definition D.1), in which case $K_U(\omega, A) = \mathbb{P}_{\mathcal{H}_U}(\omega, A)$; secondly, if $\mathcal{H}_U$ has no causal effect on $A$ (see Definition B.1), in which case $K_U(\omega, A) = \mathbb{P}(A)$; and finally, if $A \in \mathcal{H}_U$, in which case, by intervention determinism (Definition 2.2(ii)), we have $K_U(\omega, A) = 1_A(\omega)$. In the latter case, we do not even have dependence on $\mathbb{P}$. Can we generalise these results?

**Lemma D.3.** *Let* $(\Omega, \mathcal{H}, \mathbb{P}, \mathbb{K}) = (\times_{t \in T} E_t, \otimes_{t \in T} \mathcal{E}_t, \mathbb{P}, \mathbb{K})$ *be a causal space. Let* $A \in \mathcal{H}$ *be an event, and* $U \in \mathcal{P}(T)$. *If there exists a sub-$\sigma$-algebra* $\mathcal{G}$ *of* $\mathcal{H}$ *(not necessarily of the form* $\mathcal{H}_V$ *for some* $V \in \mathcal{P}(T)$*) such that*

    *(i)* *the conditional probability* $\mathbb{P}^{\mathrm{do}(U,\mathbb{Q})}_{\mathcal{H}_U \vee \mathcal{G}}(\cdot, A)$ *can be written in terms of* $\mathbb{P}$ *and* $\mathbb{Q}$*;*

    *(ii)* *the causal kernel* $K_U(\cdot, B)$ *can be written in terms of* $\mathbb{P}$ *for all* $B \in \mathcal{G}$*;*

*then* $\mathbb{P}^{\mathrm{do}(U,\mathbb{Q})}(A)$ *can be written in terms of* $\mathbb{P}$ *and* $\mathbb{Q}$.

**Remark D.4.** The three cases discussed in the paragraph above Lemma D.3 are special cases of the Lemma with $\mathcal{G}$ being any sub-$\sigma$-algebra of $\mathcal{H}$ with $\{\emptyset, \Omega\} \subseteq \mathcal{G} \subseteq \mathcal{H}_U$. In this case, condition (ii) is trivially satisfied since we have $K_U(\cdot, B) = 1_B(\cdot)$ by intervention determinism (Definition 2.2(ii)), and for condition (i), by Theorem D.2(i), we have $\mathbb{P}^{\mathrm{do}(U,\mathbb{Q})}_{\mathcal{H}_U}(\cdot, A) = K_U(\cdot, A)$, which means that the problem reduces to checking if $K_U(\cdot, A)$ can be written in terms of $\mathbb{P}$.

*Proof.* By law of total expectations, for any $V \in \mathcal{P}(T)$, we have

$$\mathbb{P}^{\mathrm{do}(U,\mathbb{Q})}(A) = \int \mathbb{P}^{\mathrm{do}(U,\mathbb{Q})}_{\mathcal{H}_U \vee \mathcal{G}}(\omega, A) \mathbb{P}^{\mathrm{do}(U,\mathbb{Q})}(d\omega)$$

$$= \int \mathbb{P}^{\mathrm{do}(U,\mathbb{Q})}_{\mathcal{H}_U \vee \mathcal{G}}(\omega, A) \int \mathbb{Q}(d\omega') K_U(\omega', d\omega).$$

Here, $\mathbb{P}^{\mathrm{do}(U,\mathbb{Q})}_{\mathcal{H}_U \vee \mathcal{G}}(\omega, A)$ can be written in terms of $\mathbb{P}$ and $\mathbb{Q}$ by condition (i). Moreover, note that it suffices to be able to write the restriction of $K_U(\omega', \cdot)$ to $\mathcal{H}_U \vee \mathcal{G}$ in terms of $\mathbb{P}$, since the integration is of a $\mathcal{H}_U \vee \mathcal{G}$-measurable function. Since the collection of intersections $\{D \cap B, D \in \mathcal{H}_U, B \in \mathcal{G}\}$ is a $\pi$-system that generates $\mathcal{H}_U \vee \mathcal{G}$ [14, p.5, 1.18], it suffices to check that $K_U(\omega', D \cap B)$ can be written in terms of $\mathbb{P}$ for all $D \in \mathcal{H}_U$ and $B \in \mathcal{G}$. But by interventional determinism (Definition 2.2(ii)), we have $K_U(\omega', D \cap B) = 1_D(\omega') K_U(\omega', B)$. Since $K_U(\omega', B)$ can be written in terms of $\mathbb{P}$ by condition (ii), the restriction of $K_U(\omega', \cdot)$ to $\mathcal{H}_U \vee \mathcal{G}$ can be written in terms of $\mathbb{P}$, and hence $\mathbb{P}^{\mathrm{do}(U,\mathbb{Q})}(A)$ can be written in terms of $\mathbb{P}$ and $\mathbb{Q}$. $\square$

**Corollary D.5.** *Let* $(\Omega, \mathcal{H}, \mathbb{P}, \mathbb{K}) = (\times_{t \in T} E_t, \otimes_{t \in T} \mathcal{E}_t, \mathbb{P}, \mathbb{K})$ *be a causal space. Let* $A \in \mathcal{H}$ *be an event, and* $U \in \mathcal{P}(T)$. *If there exists a* $V \in \mathcal{P}(T)$ *such that condition (i) of Lemma D.3 is satisfied with* $\mathcal{G} = \mathcal{H}_V$ *and one of the following conditions is satisfied:*

    *(a)* $\mathcal{H}_U$ *is a local source of* $\mathcal{H}_V$*; or*

    *(b)* $\mathcal{H}_U$ *has no causal effect on* $\mathcal{H}_V$*; or*

    *(c)* $V \subseteq U$,

*then* $\mathbb{P}^{\mathrm{do}(U,\mathbb{Q})}(A)$ *can be written in terms of* $\mathbb{P}$ *and* $\mathbb{Q}$.

*Proof.* Condition (i) of Lemma D.3 is satisfied by hypothesis. If one of (a), (b) or (c) is satisfied, then trivially, condition (ii) of Lemma D.3 is also satisfied. The result now follows from Lemma D.3. $\square$

The above is reminiscent of "valid adjustments" in the context of structural causal models [46, p.115, Proposition 6.41], and in fact contains the valid adjustments.

# E Counterfactuals

There are various notions of counterfactuals in the literature. The one considered in the SCM literature is the *interventional counterfactual*, which captures the notion of "what would have happened if we intervened on the space, given some observations (that are possibly contradictory to the intervention we imagine we would have done)". Recently, *backtracking counterfactuals* have also been integrated into the SCM framework [58]. This captures the notion of "what would have happened if background conditions of the world had been different, given that the causal laws of the system stay the same?" Finally, we note that in the potential outcomes framework, the random variables representing "potential outcomes" that form the primitives of the framework can be directly counterfactual.

Vanilla probability measures have just one argument, i.e. the event. Conditional measures and causal kernels (in the sense of our Definition 2.2) have two arguments, the first being the outcome which we either observe or force the occurrence of, and the second being the event in whose measure we are interested. For both of the above concepts of counterfactuals, we need to go one step further and consider three arguments. The first is the outcome which we observe, just like in conditioning, and the last should be the event in whose measure we are interested. For interventional counterfactuals, the second argument should be an outcome which we imagine to have forced the occurrence of given that we observed the outcome of the first argument, and for backtracking counterfactuals, the second argument should be an outcome which we imagine to have observed instead of the outcome in the first argument which we actually observed.

From these principles, we tentatively propose to extend Definition 2.2 to account for *interventional counterfactuals* as follows.

**Definition E.1.** A *causal space* is defined as the quadruple $(\Omega, \mathcal{H}, \mathbb{P}, \mathbb{K})$, where $(\Omega, \mathcal{H}, \mathbb{P}) = (\times_{t \in T} E_t, \otimes_{t \in T} \mathcal{E}_t, \mathbb{P})$ is a probability space and $\mathbb{K} = \{K_{S,\mathcal{F}} : S \in \mathcal{P}(T), \mathcal{F} \text{ sub-}\sigma\text{-algebra of } \mathcal{H}\}$, called the *causal mechanism*, is a collection of functions $K_{S,\mathcal{F}} : \Omega \times \Omega \times \mathcal{H} \to [0, 1]$, called the *causal kernel on $\mathcal{H}_S$ after observing $\mathcal{F}$*, such that

(i) for each fixed $\eta \in \Omega$ and $A \in \mathcal{H}$, $K_{S,\mathcal{F}}(\cdot, \eta, A)$ is measurable with respect to $\mathcal{H}_S$;

(ii) for each fixed $\omega \in \Omega$ and $A \in \mathcal{H}$, $K_{S,\mathcal{F}}(\omega, \cdot, A)$ is measurable with respect to $\mathcal{F}$;

(iii) for each fixed pair $(\omega, \eta) \in \Omega \times \Omega$, $K_{S,\mathcal{F}}(\omega, \eta, \cdot)$ is a measure on $\mathcal{H}$;

(iv) for all $A \in \mathcal{H}$ and $\omega, \eta \in \Omega$,
$$K_{\emptyset, \mathcal{F}}(\omega, \eta, A) = \mathbb{P}_\mathcal{F}(\eta, A);$$

(v) for all $A \in \mathcal{H}_S$, all $B \in \mathcal{H}$ and all $\omega, \eta \in \Omega$,
$$K_{S,\mathcal{F}}(\omega, \eta, A \cap B) = 1_A(\omega) K_S(\omega, \eta, B);$$
in particular, for $A \in \mathcal{H}_S$, $K_{S,\mathcal{F}}(\omega, \eta, A) = 1_A(\omega) K_{S,\mathcal{F}}(\omega, \eta, \Omega) = 1_A(\omega)$;

(vi) for all $A \in \mathcal{H}$, $\omega \in \Omega$ and sub-$\sigma$-algebras $\mathcal{F} \subseteq \mathcal{G} \subseteq \mathcal{H}$,
$$\mathbb{E}_\mathcal{F}\left[K_{S,\mathcal{G}}(\omega, \cdot, A)\right] = K_{S,\mathcal{F}}(\omega, \cdot, A).$$

Note that letting $\mathcal{F} = \{\emptyset, \Omega\}$ trivially recovers the causal space as defined in Definition 2.2. Moreover, letting $S = \emptyset$, we recover the conditional distribution given $\mathcal{F}$.

Recall that the one of the biggest philosophical differences between the SCM framework and our proposed causal spaces (Definition 2.2) was the fact that SCMs start with the variables, the structural equations and the noise distributions as the *primitive objects*, and the observational and interventional distributions over the endogenous variables are *derived* from these, whereas causal spaces take the observational and interventional distributions as the *primitive objects* (the latter via causal kernels). Note that, in the above extended definition of causal spaces incorporating interventional counterfactuals (Definition E.1), we applied the same principles, in that we treated the observational distribution ($\mathbb{P}$), interventional distributions ($K_{S,\{\emptyset,\Omega\}}$) and the (interventional) counterfactual distributions ($K_{S,\mathcal{F}}$) as the primitive objects.

This differs significantly from the SCM framework, where again, the (interventional) counterfactual distributions are *derived* from the structural equations, by first conditioning on the observed values of

the endogenous variables to get a modified (often Dirac) measure on the exogenous variables, then intervening on some of the endogenous variables, deriving the measure on the rest of the endogenous variables by propagating these through the same structural equations. We see the value in this approach in that the (interventional) counterfactual distributions can be neatly derived from the same primitive objects that are used to calculate the observational and interventional distribution. However, we argue that this cannot be an *axiomatisation* of (interventional) counterfactual distributions in the strictest sense, because it relies on assumptions. In particular, it strongly relies on the assumption that the endogenous variables have no causal effect on the exogenous variables, and when this assumption is violated, i.e. when there is a hidden mediator, calculation of (interventional) counterfactual distributions is not possible. In contrast, Definition E.1 treat the (interventional) counterfactual measures as the primitive objects, and does not impose any a priori assumptions about the system.

As mentioned in Section 5 of the main body of the paper, we leave further developments of this interventional counterfactual causal space, as well as the definition of backtracking counterfactual causal space, as essential future work.

## F    Proofs

**Theorem 2.6.** From Definition 2.3, $\mathbb{P}^{do(U,\mathbb{Q})}$ is indeed a measure on $(\Omega, \mathcal{H})$, and $\mathbb{K}^{do(U,\mathbb{Q},\mathbb{L})}$ is indeed a valid causal mechanism on $(\Omega, \mathcal{H}, \mathbb{P}^{do(U,\mathbb{Q})})$, i.e. they satisfy the axioms of Definition 2.2.

*Proof.* That $\mathbb{P}^{do(U,\mathbb{Q})}$ is a measure on $(\Omega, \mathcal{H})$ follows immediately from the usual construction of measures from measures and transition probability kernels, see e.g. Çınlar [14, p.38, Theorem 6.3]. It remains to check that $\mathbb{K}^{do(U,\mathbb{Q},\mathbb{L})}$ is a valid causal mechanism in the sense of Definition 2.2.

(i) For all $A \in \mathcal{H}$ and $\omega \in \Omega$,

$$
\begin{aligned}
K_\emptyset^{do(U,\mathbb{Q},\mathbb{L})}(\omega, A) &= \int L_\emptyset(\omega_\emptyset, d\omega_U') K_U((\omega_\emptyset, \omega_U'), A) \\
&= \int \mathbb{Q}(d\omega') K_U(\omega', A) \\
&= \mathbb{P}^{do(U,\mathbb{Q})}(A),
\end{aligned}
$$

where we applied Axiom 2.2(i) to $L_\emptyset$.

(ii) For all $A \in \mathcal{H}_S$ and $B \in \mathcal{H}$, we have, by Axiom 2.2(ii) using the fact that $A \in \mathcal{H}_S \subseteq \mathcal{H}_{S \cup U}$,

$$
\begin{aligned}
&K_S^{do(U,\mathbb{Q},\mathbb{L})}(\omega, A \cap B) \\
&= \int L_{S \cap U}(\omega_{S \cap U}, d\omega_U') K_{S \cup U}((\omega_{S \setminus U}, \omega_U'), A \cap B) \\
&= \int L_{S \cap U}(\omega_{S \cap U}, d\omega_U') 1_A((\omega_{S \setminus U}, \omega_U')) K_{S \cup U}((\omega_{S \setminus U}, \omega_U'), B) \\
&= \int L_{S \cap U}(\omega_{S \cap U}, d\omega_U') 1_A((\omega_{S \setminus U}, \omega_{S \cap U}')) K_{S \cup U}((\omega_{S \setminus U}, \omega_U'), B),
\end{aligned}
$$

where, in going from the third line to the fourth, we split the $\omega_U'$ in $1_A((\omega_{S \setminus U}, \omega_U'))$ into components $(\omega_{S \cap U}', \omega_{U \setminus S}')$ and notice that since $A \in \mathcal{H}_S$, $1_A$ does not depend on the component $\omega_{U \setminus S}'$. Here, the map $\omega_{S \cap U}' \mapsto 1_A((\omega_{S \setminus U}, \omega_{S \cap U}'))$ is $\mathcal{H}_{S \cap U}$-measurable, so we can write it as the limit of an increasing sequence of positive $\mathcal{H}_{S \cap U}$-simple functions (see Section A.1), say $(f_n)_{n \in \mathbb{N}}$ with $f_n = \sum_{i_n=1}^{m_n} b_{i_n} 1_{B_{i_n}}$, where $B_{i_n} \in \mathcal{H}_{S \cap U}$. Likewise, the map $\omega_U' \mapsto K_{S \cup U}((\omega_{S \setminus U}, \omega_U'), B)$ is $\mathcal{H}_U$-measurable, so we can write it as the limit of an increasing sequence of positive $\mathcal{H}_U$-simple functions, say $(g_n)_{n \in \mathbb{N}}$ with $g_n = \sum_{j_n=1}^{l_n} c_{j_n} 1_{C_{j_n}}$, where $C_{j_n} \in \mathcal{H}_U$. Hence

$$
K_S^{do(U,\mathbb{Q},\mathbb{L})}(\omega, A \cap B) = \int L_{S \cap U}(\omega_{S \cap U}, d\omega_U') \left( \lim_{n \to \infty} f_n(\omega_{S \cap U}') \right) \left( \lim_{n \to \infty} g_n(\omega_U') \right).
$$

Since, for each $\omega'_U$, both of the limits exist by construction, namely the original measurable functions, we have that the product of the limits is the limit of the products:

$$K_S^{\text{do}(U,\mathbb{Q},\mathbb{L})}(\omega, A \cap B) = \int L_{S \cap U}(\omega_{S \cap U}, d\omega'_U) \lim_{n \to \infty} \left( f_n(\omega'_{S \cap U}) g_n(\omega'_U) \right).$$

Here, since $f_n$ and $g_n$ were individually sequences of increasing functions, the pointwise products $f_n g_n$ also form an increasing sequence of functions. Hence, we can apply the monotone convergence theorem to see that

$$K_S^{\text{do}(U,\mathbb{Q},\mathbb{L})}(\omega, A \cap B)$$

$$= \lim_{n \to \infty} \int L_{S \cap U}(\omega_{S \cap U}, d\omega'_U) f_n(\omega'_{S \cap U}) g_n(\omega'_U)$$

$$= \lim_{n \to \infty} \sum_{i_n=1}^{m_n} \sum_{j_n=1}^{l_n} b_{i_n} c_{j_n} \int L_{S \cap U}(\omega_{S \cap U}, d\omega'_U) 1_{B_{i_n}}(\omega'_{S \cap U}) 1_{C_{j_n}}(\omega'_U)$$

$$= \lim_{n \to \infty} \sum_{i_n=1}^{m_n} \sum_{j_n=1}^{l_n} b_{i_n} c_{j_n} L_{S \cap U}(\omega_{S \cap U}, B_{i_n} \cap C_{j_n})$$

$$= \lim_{n \to \infty} \sum_{i_n=1}^{m_n} \sum_{j_n=1}^{l_n} b_{i_n} c_{j_n} 1_{B_{i_n}}(\omega_{S \cap U}) L_{S \cap U}(\omega_{S \cap U}, C_{j_n})$$

$$= \lim_{n \to \infty} \sum_{i_n=1}^{m_n} b_{i_n} 1_{B_{i_n}}(\omega_{S \cap U}) \sum_{j_n=1}^{l_n} c_{j_n} L_{S \cap U}(\omega_{S \cap U}, C_{j_n})$$

$$= \left( \lim_{n \to \infty} \sum_{i_n=1}^{m_n} b_{i_n} 1_{B_{i_n}}(\omega_{S \cap U}) \right) \left( \lim_{n \to \infty} \sum_{j_n=1}^{l_n} c_{j_n} L_{S \cap U}(\omega_{S \cap U}, C_{j_n}) \right)$$

$$= \left( \lim_{n \to \infty} f_n(\omega_{S \cap U}) \right) \left( \lim_{n \to \infty} \int L_{S \cap U}(\omega_{S \cap U}, d\omega'_U) \sum_{j_n=1}^{l_n} c_j 1_{C_{j_n}}(\omega'_U) \right)$$

$$= 1_A((\omega_{S \setminus U}, \omega_{S \cap U})) \int L_{S \cap U}(\omega_{S \cap U}, d\omega'_U) \lim_{n \to \infty} g_n(\omega'_U)$$

$$= 1_A(\omega_S) \int L_{S \cap U}(\omega_{S \cap U}, d\omega'_U) K_{S \cup U}((\omega_{S \setminus U}, \omega'_U), B)$$

$$= 1_A(\omega_S) K_S^{\text{do}(U,\mathbb{Q},\mathbb{L})}(\omega_S, B)$$

where, from the fourth line to the fifth, we used Axiom 2.2(ii); from the sixth line to the seventh, we used that limit of the products is the product of the limits again, noting that both of the limits exist by construction; from the eighth line to the ninth, we used monotone convergence theorem again. This is the required result.

$$\square$$

**Theorem C.3.** Let $(\Omega, \mathcal{H}, \mathbb{P}, \mathbb{K}) = (\times_{t \in T} E_t, \otimes_{t \in T} \mathcal{E}_t, \mathbb{P}, \mathbb{K})$ be a causal space, and $U \in \mathcal{P}(T)$ and $\mathbb{Q}$ a probability measure on $(\Omega, \mathcal{H}_U)$. Then after a hard intervention on $\mathcal{H}_U$ via $\mathbb{Q}$, the intervention causal kernels $K_S^{\text{do}(U,\mathbb{Q},\text{hard})}$ are given by

$$K_S^{\text{do}(U,\mathbb{Q},\text{hard})}(\omega, A) = K_S^{\text{do}(U,\mathbb{Q},\text{hard})}(\omega_S, A) = \int \mathbb{Q}(d\omega'_{U \setminus S}) K_{S \cup U}((\omega_S, \omega'_{U \setminus S}), A).$$

*Proof.* We decompose $\mathcal{H}_U$ as a product $\sigma$-algebra into $\mathcal{H}_{S \cap U} \otimes \mathcal{H}_{U \setminus S}$. Then events of the form $B \cap C$ with $B \in \mathcal{H}_{S \cap U}$ and $C \in \mathcal{H}_{U \setminus S}$ generate $\mathcal{H}_U$, so for fixed $\omega_{S \cap U}$, the measure $L_{S \cap U}(\omega_{S \cap U}, \cdot)$ is completely determined by $L_{S \cap U}(\omega_{S \cap U}, B \cap C)$ for all $B \in \mathcal{H}_{S \cap U}$, $C \in \mathcal{H}_{U \setminus S}$. But we have

$$L_{S \cap U}(\omega_{S \cap U}, B \cap C) = \delta_{\omega_{S \cap U}}(B) L_{S \cap U}(\omega_{S \cap U}, C) \qquad \text{by Axiom 2.2(ii)}$$

$$= \delta_{\omega_{S \cap U}}(B) \mathbb{Q}(C),$$

since $L_{S \cap U}$ is trivial and $C \in \mathcal{H}_{U \setminus S}$. So the measure $L_{S \cap U}(\omega_{S \cap U}, \cdot)$ is a product measure of $\delta_{\omega_{S \cap U}}$ and $\mathbb{Q}$. Hence, applying Fubini's theorem,

$$
\begin{aligned}
K_S^{\text{do}(U, \mathbb{Q}, \text{hard})}(\omega, A) &= \int L_{S \cap U}(\omega_{S \cap U}, d\omega'_U) K_{S \cup U}((\omega_{S \setminus U}, \omega'_U), A) \\
&= \int \int K_{S \cup U}((\omega_{S \setminus U}, \omega'_{S \cap U}, \omega'_{U \setminus S}), A) \delta_{\omega_{S \cap U}}(d\omega'_{S \cap U}) \mathbb{Q}(d\omega'_{U \setminus S}) \\
&= \int K_{S \cup U}((\omega_{S \setminus U}, \omega_{S \cap U}, \omega'_{U \setminus S}), A) \mathbb{Q}(d\omega'_{U \setminus S}) \\
&= \int \mathbb{Q}(d\omega'_{U \setminus S}) K_{S \cup U}((\omega_S, \omega'_{U \setminus S}), A),
\end{aligned}
$$

as required.

$\square$

**Theorem D.2.** Suppose we have a causal space $(\Omega, \mathcal{H}, \mathbb{P}, \mathbb{K}) = (\times_{t \in T} E_t, \otimes_{t \in T} \mathcal{E}_t, \mathbb{P}, \mathbb{K})$, and let $U \in \mathcal{P}(T)$.

- (i) For any measure $\mathbb{Q}$ on $\mathcal{H}_U$ and any causal mechanism $\mathbb{L}$ on $(\Omega, \mathcal{H}_U, \mathbb{Q})$, the causal kernel $K_U^{\text{do}(U, \mathbb{Q}, \mathbb{L})} = K_U$ is a version of $\mathbb{P}_{\mathcal{H}_U}^{\text{do}(U, \mathbb{Q})}$, which means that $\mathcal{H}_U$ is a global source $\sigma$-algebra of the intervened causal space $(\Omega, \mathcal{H}, \mathbb{P}^{\text{do}(U, \mathbb{Q})}, \mathbb{K}^{\text{do}(U, \mathbb{Q}, \mathbb{L})})$.

- (ii) Suppose $V \in \mathcal{P}(T)$ with $V \subseteq U$. Suppose that the measure $\mathbb{Q}$ on $(\Omega, \mathcal{H}_U)$ factorises over $\mathcal{H}_V$ and $\mathcal{H}_{U \setminus V}$, i.e. for any $A \in \mathcal{H}_V$ and $B \in \mathcal{H}_{U \setminus V}$, $\mathbb{Q}(A \cap B) = \mathbb{Q}(A)\mathbb{Q}(B)$. Then after a hard intervention on $\mathcal{H}_U$ via $\mathbb{Q}$, the causal kernel $K_V^{\text{do}(U, \mathbb{Q})}$ is a version of $\mathbb{P}_V^{\text{do}(U, \mathbb{Q})}$, which means that $\mathcal{H}_V$ is a global source $\sigma$-algebra of the intervened causal space $(\Omega, \mathcal{H}, \mathbb{P}^{\text{do}(U, \mathbb{Q})}, \mathbb{K}^{\text{do}(U, \mathbb{Q})})$.

*Proof.* Suppose that $f = \sum_{i=1}^m b_i 1_{B_i}$ is a $\mathcal{H}_U$-simple function, i.e. with $B_i \in \mathcal{H}_U$ for $i = 1, ..., m$. Then for any $B \in \mathcal{H}_U$,

$$
\begin{aligned}
\int_B f(\omega) \mathbb{P}^{\text{do}(U, \mathbb{Q})}(d\omega) &= \int_B \sum_{i=1}^m b_i 1_{B_i}(\omega) \mathbb{P}^{\text{do}(U, \mathbb{Q})}(d\omega) \\
&= \sum_{i=1}^m b_i \mathbb{P}^{\text{do}(U, \mathbb{Q})}(B \cap B_i) \\
&= \sum_{i=1}^m b_i \int \mathbb{Q}(d\omega) K_U(\omega, B \cap B_i) \quad \text{by the definition of } \mathbb{P}^{\text{do}(U, \mathbb{Q})} \\
&= \sum_{i=1}^m b_i \int \mathbb{Q}(d\omega) 1_{B \cap B_i}(\omega) \quad\quad \text{by Axiom 2.2(ii)} \\
&= \int_B \sum_{i=1}^m b_i 1_{B_i}(\omega) \mathbb{Q}(d\omega) \\
&= \int_B f(\omega) \mathbb{Q}(d\omega).
\end{aligned}
$$

Now, for any $\mathcal{H}_U$-measurable map $g : \Omega \to \mathbb{R}$, we can write it as a limit of an increasing sequence of positive $\mathcal{H}_U$-simple functions $f_n$ (see Section A.1), so for any $B \in \mathcal{H}_U$,

$$
\begin{aligned}
\int_B g(\omega) \mathbb{P}^{\text{do}(U, \mathbb{Q})}(d\omega) &= \int_B \lim_{n \to \infty} f_n(\omega) \mathbb{P}^{\text{do}(U, \mathbb{Q})}(d\omega) \\
&= \lim_{n \to \infty} \int_B f_n(\omega) \mathbb{P}^{\text{do}(U, \mathbb{Q})}(d\omega) \quad\quad \text{by the monotone convergence theorem}
\end{aligned}
$$

$$
\begin{aligned}
&= \lim_{n\to\infty} \int_B f_n(\omega)\mathbb{Q}(d\omega) && \text{by above}\\
&= \int_B \lim_{n\to\infty} f_n(\omega)\mathbb{Q}(d\omega) && \text{by the monotone convergence theorem}\\
&= \int_B g(\omega)\mathbb{Q}(d\omega).
\end{aligned}
$$

We use this fact in the proof of both parts of this theorem.

(i) First note that we indeed have $K_U^{\mathrm{do}(U,\mathbb{Q},\mathbb{L})} = K_U$, by Remark C.1(a). For any $A \in \mathcal{H}$, the map $\omega \mapsto K_U(\omega, A)$ is $\mathcal{H}_U$-measurable, so for any $B \in \mathcal{H}_U$,

$$
\begin{aligned}
\int_B K_U(\omega, A)\mathbb{P}^{\mathrm{do}(U,\mathbb{Q})}(d\omega) &= \int_B K_U(\omega, A)\mathbb{Q}(d\omega) && \text{by above fact}\\
&= \int 1_B(\omega)K_U(\omega, A)\mathbb{Q}(d\omega)\\
&= \int K_U(\omega, A\cap B)\mathbb{Q}(d\omega) && \text{by Axiom 2.2(ii)}\\
&= \mathbb{P}^{\mathrm{do}(U,\mathbb{Q})}(A\cap B)\\
&= \int 1_{A\cap B}(\omega)\mathbb{P}^{\mathrm{do}(U,\mathbb{Q})}(d\omega)\\
&= \int 1_B(\omega)1_A(\omega)\mathbb{P}^{\mathrm{do}(U,\mathbb{Q})}(d\omega)\\
&= \int_B 1_A(\omega)\mathbb{P}^{\mathrm{do}(U,\mathbb{Q})}(d\omega).
\end{aligned}
$$

So $K_U(\cdot, A) = K_U^{\mathrm{do}(U,\mathbb{Q},\mathbb{L})}(\cdot, A)$ is indeed a version of the conditional probability $\mathbb{P}^{\mathrm{do}(U,\mathbb{Q})}_{\mathcal{H}_U}(A)$, which means that $\mathcal{H}_U$ is a global source of $(\Omega, \mathcal{H}, \mathbb{P}^{\mathrm{do}(U,\mathbb{Q})}, \mathbb{K}^{\mathrm{do}(U,\mathbb{Q},\mathbb{L})})$.

(ii) For any $A \in \mathcal{H}$, the map $\omega \mapsto K_V^{\mathrm{do}(U,\mathbb{Q})}(\omega, A)$ is $\mathcal{H}_V$-measurable and hence $\mathcal{H}_U$-measurable, so for any $B \in \mathcal{H}_V \subseteq \mathcal{H}_U$,

$$
\begin{aligned}
&\int_B K_V^{\mathrm{do}(U,\mathbb{Q})}(\omega_V, A)\mathbb{P}^{\mathrm{do}(U,\mathbb{Q})}(d\omega_V)\\
&= \int_B K_V^{\mathrm{do}(U,\mathbb{Q})}(\omega_V, A)\mathbb{Q}(d\omega_V) && \text{by above fact}\\
&= \int K_V^{\mathrm{do}(U,\mathbb{Q})}(\omega_V, A\cap B)\mathbb{Q}(d\omega_V) && \text{by Axiom 2.2(ii)}\\
&= \int\int \mathbb{Q}(d\omega'_{U\setminus V})K_U((\omega_V, \omega'_{U\setminus V}), A\cap B)\mathbb{Q}(d\omega_V)\\
&= \int K_U(\omega_U, A\cap B)\mathbb{Q}(d\omega_U)\\
&= \int_B 1_A(\omega)\mathbb{P}^{\mathrm{do}(U,\mathbb{Q})}(d\omega).
\end{aligned}
$$

where, in going from the third line to the fourth, we used Theorem C.3, and to go from the fourth line to the fifth, we used the hypothesis that $\mathbb{Q}$ factorises over $\mathcal{H}_V$ and $\mathcal{H}_{U\setminus V}$, meaning $\mathbb{Q}(d\omega_{U\setminus V})\mathbb{Q}(d\omega_V) = \mathbb{Q}(d\omega_U)$. So $K_V^{\mathrm{do}(U,\mathbb{Q})}(\omega, A)$ is indeed a version of the conditional probability $\mathbb{P}^{\mathrm{do}(U,\mathbb{Q})}_{\mathcal{H}_V}(A)$, which means that $\mathcal{H}_V$ is a global source of $(\Omega, \mathcal{H}, \mathbb{P}^{\mathrm{do}(U,\mathbb{Q})}, \mathbb{K}^{\mathrm{do}(U,\mathbb{Q})})$.

$\square$

**Lemma C.4.** Let $(\Omega, \mathcal{H}, \mathbb{P}, \mathbb{K}) = (\times_{t\in T} E_t, \otimes_{t\in T}\mathcal{E}_t, \mathbb{P}, \mathbb{K})$ be a causal space, and $U \in \mathcal{P}(T)$, $\mathbb{Q}$ a probability measure on $(\Omega, \mathcal{H}_U)$ and $\mathbb{L} = \{L_V : V \in \mathcal{P}(U)\}$ a causal mechanism on $(\Omega, \mathcal{H}_U, \mathbb{Q})$. Suppose we intervene on $\mathcal{H}_U$ via $(\mathbb{Q}, \mathbb{L})$.

(i) For $A \in \mathcal{H}_U$ and $V \in \mathcal{P}(T)$ with $V \cap U = \emptyset$, $\mathcal{H}_V$ has no causal effect on $A$ (c.f. Definition B.1(i)) in the intervention causal space $(\Omega, \mathcal{H}, \mathbb{P}^{\mathrm{do}(U,\mathbb{Q})}, \mathbb{K}^{\mathrm{do}(U,\mathbb{Q},\mathbb{L})})$, i.e. events in the $\sigma$-algebra $\mathcal{H}_U$ on which intervention took place are not causally affected by $\sigma$-algebras outside $\mathcal{H}_U$.

(ii) Again, let $V \in \mathcal{P}(T)$ with $V \cap U = \emptyset$, and also let $A \in \mathcal{H}$ be any event. If, in the original causal space, $\mathcal{H}_V$ had no causal effect on $A$, then in the intervention causal space, $\mathcal{H}_V$ has no causal effect on $A$ either.

(iii) Now let $V \in \mathcal{P}(T)$, $A \in \mathcal{H}$ any event and suppose that the intervention on $\mathcal{H}_U$ via $\mathbb{Q}$ is hard. Then if $\mathcal{H}_V$ had no causal effect on $A$ in the original causal space, then $\mathcal{H}_V$ has no causal effect on $A$ in the intervention causal space either.

*Proof.*   (i) Take any $S \in \mathcal{P}(T)$. See that

$$
\begin{aligned}
K_S^{\mathrm{do}(U,\mathbb{Q},\mathbb{L})}(\omega, A) &= \int L_{S \cap U}(\omega_{S \cap U}, d\omega_U') K_{S \cup U}((\omega_{S \setminus U}, \omega_U'), A) \\
&= \int L_{S \cap U}(\omega_{S \cap U}, d\omega_U') 1_A(\omega_U') \\
&= \int L_{S \cap U}(\omega_{S \cap U}, d\omega_U') K_{(S \setminus V) \cup U}((\omega_{(S \setminus V) \setminus U}, \omega_U'), A) \\
&= \int L_{(S \setminus V) \cap U}(\omega_{(S \setminus V) \cap U}, d\omega_U') K_{(S \setminus V) \cup U}((\omega_{(S \setminus V) \setminus U}, \omega_U'), A) \\
&= K_{S \setminus V}^{\mathrm{do}(U,\mathbb{Q},\mathbb{L})}(\omega, A)
\end{aligned}
$$

where, in going from the first line to the second and from the second line to the third, we used the fact that $A \in \mathcal{H}_U$, and in going from the third line to the fourth, we applied the fact that $(S \setminus V) \cap U = S \cap U$ since $V \cap U = \emptyset$. Since $S \in \mathcal{P}(T)$ was arbitrary, $\mathcal{H}_V$ has no causal effect on $A$ in the intervention causal space.

(ii) Take any $S \in \mathcal{P}(T)$. See that

$$
\begin{aligned}
K_S^{\mathrm{do}(U,\mathbb{Q},\mathbb{L})}(\omega, A) &= \int L_{S \cap U}(\omega_{S \cap U}, d\omega_U') K_{S \cup U}((\omega_{S \setminus U}, \omega_U'), A) \\
&= \int L_{S \cap U}(\omega_{S \cap U}, d\omega_U') K_{(S \cup U) \setminus V}((\omega_{(S \setminus V) \setminus U}, \omega_U'), A) \\
&= \int L_{(S \setminus V) \cap U}(\omega_{(S \setminus V) \cap U}, d\omega_U') K_{(S \setminus V) \cup U}((\omega_{(S \setminus V) \setminus U}, \omega_U'), A) \\
&= K_{S \setminus V}^{\mathrm{do}(U,\mathbb{Q},\mathbb{L})}(\omega, A)
\end{aligned}
$$

where, in going from the first line to the second, we used the fact that $\mathcal{H}_V$ has no causal effect on $A$ in the original causal space, and in going from the second line to the third, we used $U \cap V = \emptyset$, which gives us $S \cap U = (S \setminus V) \cap U$ and $(S \cup U) \setminus V = (S \setminus V) \cup U$. Since $S \in \mathcal{P}(T)$ was arbitrary, $\mathcal{H}_V$ has no causal effect on $A$ in the intervention causal space.

(iii) Take any $S \in \mathcal{P}(T)$. Apply Theorem C.3 to see that

$$
\begin{aligned}
K_S^{\mathrm{do}(U,\mathbb{Q},\mathrm{hard})}(\omega, A) &= \int \mathbb{Q}(d\omega_{U \setminus S}') K_{S \cup U}((\omega_S, \omega_{U \setminus S}'), A) \\
&= \int \mathbb{Q}(d\omega_{U \setminus S}') K_{(S \cup U) \setminus V}((\omega_S, \omega_{U \setminus S}'), A) && \text{Def. B.1(i)} \\
&= \int \mathbb{Q}(d\omega_{U \setminus S}') K_{((S \setminus V) \cup U) \setminus V}((\omega_S, \omega_{U \setminus S}'), A) \\
&= \int \mathbb{Q}(d\omega_{U \setminus S}') K_{(S \setminus V) \cup U}((\omega_S, \omega_{U \setminus S}'), A) && \text{Def. B.1(i)} \\
&= \int \mathbb{Q}(d\omega_{U \setminus (S \setminus V)}') K_{(S \setminus V) \cup U}((\omega_{S \setminus V}, \omega_{U \setminus (S \setminus V)}'), A)
\end{aligned}
$$

$$= K_{S\setminus V}^{\mathrm{do}(U,\mathbb{Q})}(\omega, A),$$

where, in going from the second line to the third, we used that $(S\cup U)\setminus V = ((S\setminus V)\cup U)\setminus V$. Since $S \in \mathcal{P}(T)$ was arbitrary, $\mathcal{H}_V$ has no causal effect on $A$ in the intervention causal space.

$\square$

**Lemma C.5.** Let $(\Omega, \mathcal{H}, \mathbb{P}, \mathbb{K}) = (\times_{t\in T} E_t, \otimes_{t\in T}\mathcal{E}_t, \mathbb{P}, \mathbb{K})$ be a causal space, and $U \in \mathcal{P}(T)$. For an event $A \in \mathcal{H}$, if $\mathcal{H}_U$ has a dormant causal effect on $A$ in the original causal space, then there exists a hard intervention and a subset $V \subseteq U$ such that in the intervention causal space, $\mathcal{H}_V$ has an active causal effect on $A$.

*Proof.* That $\mathcal{H}_U$ has a dormant causal effect on $A$ tells us that $K_U(\omega, A) = \mathbb{P}(A)$ for all $\omega \in \Omega$, but there exists some $S \in \mathcal{P}(T)$ and some $\omega_0 \in \Omega$ such that $K_S(\omega_0, A) \neq K_{S\setminus U}(\omega_0, A)$. We must have $S \cap U \neq \emptyset$, since otherwise $S \setminus U = S$ and we cannot possibly have $K_S(\omega_0, A) \neq K_{S\setminus U}(\omega_0, A)$. Then we hard-intervene on $\mathcal{H}_{S\setminus U}$ with the Dirac measure on $\omega_0$. Then apply Theorem C.3 to see that

$$K_{S\cap U}^{\mathrm{do}(S\setminus U, \delta_{\omega_0}, \mathrm{hard})}((\omega_0)_{U\cap S}, A) = \int \delta_{\omega_0}(d\omega'_{S\setminus U}) K_S(((\omega_0)_{U\cap S}, \omega'_{S\setminus U}), A)$$
$$= K_S(\omega_0, A)$$
$$\neq K_{S\setminus U}(\omega_0, A)$$

Note that the intervention measure on $A$ is equal to $K_{S\setminus U}(\omega_0, A)$:

$$\mathbb{P}^{\mathrm{do}(S\setminus U, \delta_{\omega_0})}(A) = \int \delta_{\omega_0}(d\omega'_{S\setminus U}) K_{S\setminus U}(\omega', A) = K_{S\setminus U}(\omega_0, A).$$

Putting these together, we have

$$K_{S\cap U}^{\mathrm{do}(S\setminus U, \delta_{\omega_0}, \mathrm{hard})}(\omega_0, A) \neq \mathbb{P}^{\mathrm{do}(S\setminus U, \delta_{\omega_0})}(A),$$

i.e. in the intervention causal space $(\Omega, \mathcal{H}, \mathbb{P}^{\mathrm{do}(S\setminus U, \delta_{\omega_0})}, K_{S\cap U}^{\mathrm{do}(S\setminus U, \delta_{\omega_0}, \mathrm{hard})})$, $\mathcal{H}_{S\cap U}$ has an active causal effect on $A$. $\square$

**Lemma C.6.** Let $(\Omega, \mathcal{H}, \mathbb{P}, \mathbb{K}) = (\times_{t\in T} E_t, \otimes_{t\in T}\mathcal{E}_t, \mathbb{P}, \mathbb{K})$ be a causal space, and $U, V \in \mathcal{P}(T)$. For an event $A \in \mathcal{H}$, suppose that $\mathcal{H}_U$ has no causal effect on $A$ given $\mathcal{H}_V$ (see Definition B.4). Then after an intervention on $\mathcal{H}_V$ via any $(\mathbb{Q}, \mathbb{L})$, $\mathcal{H}_{U\setminus V}$ has no causal effect on $A$.

*Proof.* Take any probability measure $\mathbb{Q}$ on $(\Omega, \mathcal{H}_V)$ and any causal mechanism $\mathbb{L}$ on $(\Omega, \mathcal{H}_V, \mathbb{Q})$. Then see that, for any $S \in \mathcal{P}(T)$ and all $\omega \in \Omega$,

$$K_S^{\mathrm{do}(V,\mathbb{Q},\mathbb{L})}(\omega, A) = \int L_{S\cap V}(\omega_{S\cap V}, d\omega'_V) K_{S\cup V}((\omega_{S\setminus V}, \omega'_V), A)$$
$$= \int L_{S\cap V}(\omega_{S\cap V}, d\omega'_V) K_{(S\cup V)\setminus(U\setminus V)}((\omega_{S\setminus(U\cup V)}, \omega'_V), A)$$
$$= \int L_{(S\setminus(U\setminus V))\cap V}(\omega_{(S\setminus(U\setminus V))\cap V}, d\omega'_V) K_{(S\setminus(U\setminus V))\cup V}((\omega_{S\setminus(U\cup V)}, \omega'_V), A)$$
$$= K_{S\setminus(U\setminus V)}^{\mathrm{do}(V,\mathbb{Q},\mathbb{L})}(\omega, A),$$

where, in going from the first line to the second, we used the fact that $\mathcal{H}_U$ has no causal effect on $A$ given $\mathcal{H}_V$, and in going from the second line to the third, we used identities $S\cap V = (S\setminus(U\setminus V))\cap V$ and $(S\cup V)\setminus(U\setminus V) = (S\setminus(U\setminus V))\cup V$. Since $S \in \mathcal{P}(T)$ was arbitrary, we have that $\mathcal{H}_{U\setminus V}$ has no causal effect on $A$ in the intervention causal space. $\square$

**Theorem C.7.** Let $(\Omega, \mathcal{H}, \mathbb{P}, \mathbb{K}) = (\times_{t\in T} E_t, \otimes_{t\in T}\mathcal{E}_t, \mathbb{P}, \mathbb{K})$ be a causal space, where the index set $T$ can be written as $T = W \times \tilde{T}$, with $W$ representing time and $\mathbb{K}$ respecting time. Take any $U \in \mathcal{P}(T)$ and any probability measure $\mathbb{Q}$ on $\mathcal{H}_U$. Then the intervention causal mechanism $\mathbb{K}^{\mathrm{do}(U,\mathbb{Q},\mathrm{hard})}$ also respects time.

*Proof.* Take any $w_1, w_@ \in W$ with $w_1 < w_2$. Since $\mathbb{K}$ respects time, we have that $\mathcal{H}_{w_2 \times \tilde{T}}$ has no causal effect on $\mathcal{H}_{w_1 \times \tilde{T}}$ in the original causal space. To show that $\mathcal{H}_{w_2 \times \tilde{T}}$ has no causal effect on $\mathcal{H}_{w_1 \times \tilde{T}}$ after a hard intervention on $\mathcal{H}_U$ via $\mathbb{Q}$, take any $S \in \mathcal{P}(T)$ and any event $A \in \mathcal{H}_{w_1 \times \tilde{T}}$. Then using Theorem C.3,

$$K_S^{\mathrm{do}(U,\mathbb{Q},\mathrm{hard})}(\omega, A)$$

$$= \int \mathbb{Q}(d\omega') K_{S \cup U}((\omega_S, \omega'_{U \setminus S}), A)$$

$$= \int \mathbb{Q}(d\omega') K_{(S \cup U) \setminus \mathcal{H}_{w_2 \times \tilde{T}}}((\omega_{S \setminus \mathcal{H}_{w_2 \times \tilde{T}}}, \omega'_{U \setminus (S \cup \mathcal{H}_{w_2 \times \tilde{T}})}), A)$$

$$= \int \mathbb{Q}(d\omega') K_{((S \cup U) \setminus \mathcal{H}_{w_2 \times \tilde{T}}) \cup (U \cap \mathcal{H}_{w_2 \times \tilde{T}})}((\omega_{S \setminus \mathcal{H}_{w_2 \times \tilde{T}}}, \omega'_{(U \setminus (S \cup \mathcal{H}_{w_2 \times \tilde{T}})) \cup (U \cap \mathcal{H}_{w_2 \times \tilde{T}})}), A)$$

$$= \int \mathbb{Q}(d\omega') K_{(S \setminus \mathcal{H}_{w_2 \times \tilde{T}}) \cup U}((\omega_{S \setminus \mathcal{H}_{w_2 \times \tilde{T}}}, \omega'_{U \setminus (S \setminus \mathcal{H}_{w_2 \times \tilde{T}})}), A)$$

$$= K_{S \setminus \mathcal{H}_{w_2 \times \tilde{T}}}^{\mathrm{do}(U,\mathbb{Q},\mathrm{hard})}(\omega, A)$$

where, from the second line to the third, we used the fact that $\mathcal{H}_{w_2 \times \tilde{T}}$ has no causal effect on $A$, from the third line to the fourth we used the fact that $U \cap \mathcal{H}_{w_2 \times \tilde{T}}$ has no causal effect on $A$ (by Remark B.2(e)) and Remark B.2(g), and from the fourth line to the fifth, we used that $((S \cup U) \setminus \mathcal{H}_{w_2 \times \tilde{T}}) \cup (U \cap \mathcal{H}_{w_2 \times \tilde{T}}) = (S \setminus \mathcal{H}_{w_2 \times \tilde{T}}) \cup U$ and $(U \setminus (S \cup \mathcal{H}_{w_2 \times \tilde{T}})) \cup (U \cap \mathcal{H}_{w_2 \times \tilde{T}}) = U \setminus (S \setminus \mathcal{H}_{w_2 \times \tilde{T}})$. Since $S \in \mathcal{P}(T)$ was arbitrary, we have that $\mathcal{H}_{w_2 \times \tilde{T}}$ has no causal effect on $A$ (Definition B.1(i)). Since $A \in \mathcal{H}_{w_1 \times \tilde{T}}$ was arbitrary, $\mathcal{H}_{w_2 \times \tilde{T}}$ has no causal effect on $\mathcal{H}_{w_1 \times \tilde{T}}$, and so $\mathbb{K}^{\mathrm{do}(U,\mathbb{Q},\mathrm{hard})}$ respects time. $\square$

