# OpenReview forum: "A Measure-Theoretic Axiomatisation of Causality"
_NeurIPS.cc/2023/Conference — NeurIPS 2023 oral_

### Official Review · Reviewer_kUcy · 2023-06-11

**Soundness:** 4 excellent
**Presentation:** 4 excellent
**Contribution:** 4 excellent
**Rating:** 8
**Confidence:** 3

**Summary:**

This paper builds a rigorous foundation for causal reasoning. It does so by defining causal space, an extension of the concept of probability space by adding stochastic kernels. This richer structure allows the authors to define notions like interventions and causal effects, and generalize the standard frameworks like structural causal models.

**Strengths:**

It is easy to take for granted the advantages provided by the measure-theoretic foundations of probability theory, but it is undeniable that Kolmogorov's insight to do so put probability theory in a respectable position and more importantly allowed for rapid progress on otherwise impossibly difficult problems related to stochastic process by Doob and others. No doubt this axiomatization did not happen without resistance, and many early probabilists considered it an offense to probabilistic intuition.

This paper is a valiant attempt at axiomatization of causal reasoning, and in my opinion it has a done a fairly good job. The idea of using stochastic kernels to enrich the underlying probabilistic structure for the much more difficult (than plain statistics) problem of causal reasoning is very natural while at the same time difficult to effectuate. The authors do a great job defining all the relevant terms and being rigorous in their treatment.

**Weaknesses:**

The authors do a lackluster job in providing enough motivation and examples, and discussing potential alternative axiomatizations. A first read of the paper gives an impression of pulling the definitions "out of the hat". This is unfortunate since it is important to encourage as many readers as possible to think in this direction and incorporate it in their research.

On lines 137-138, it is stated that "$K_S(\omega, A) = K_S((\omega_S, \omega_{T \setminus S}), A)$ for any $A \in \mathcal{H}$ only depends on the first $\omega_S$ component of $\omega$". I don't think it is true (if it is indeed true, please provide a proof). For example, let $T = \\{1,2\\}, S = \\{1\\}, E_1 = E_2 = \\{0,1\\}$ and $\mathcal{E}_1 = \mathcal{E}_2 = \mathcal{P}(\\{0,1\\})$. Then $\Omega = E_1 \times E_2$ and $\mathcal{H} = \mathcal{E}_1 \otimes \mathcal{E}_2 = \mathcal{P}(\Omega)$. Now we could define $K_S$ in such a manner that it depends on the second component of $\omega \in \Omega$. Say:
$$
K(\omega, A) = \begin{cases}
0 &\text{if } \omega = (0,0), A = \\{(0,0)\\}\\\\
1 &\text{if } \omega = (0,1), A = \\{(0,0)\\}\\\\
\vdots
\end{cases}
$$

A few minor nitpickings:
1. In section 2, where $\mathcal{H}_S$ is defined it will be clearer to remind what measurable rectangles are, since it is implicitly being used that for measurable rectangles $A_t$ differs from $\mathcal{E}_t$ for only finite $t$'s.
2. In definition 2.2, it is misleading to say "(product) probability space". The word "product" when talking about measurable spaces (as opposed to _measure_ spaces) has a connotation of product measures, which is not the case here.
3. I find the notation in equation (2) confusing. It might be helpful to also say that the kernel $K_S^{\text{do}(U, \mathbb Q, \mathbb L)}$ is simply the product $L_{S \cap U} K_{S \cup U}$ of stochastic kernels (as defined by Çinlar on page 39).

**Questions:**

While understanding the definition of causal space, a natural question of what are some sufficient conditions for its existence arises (similar in spirit to the sufficient conditions for the existence of regular conditional distributions). It would be nice to explore this question.

**Limitations:**

Not relevant.

---

> ### Author Rebuttal · Authors · 2023-08-05
>
> We thank the reviewer very much for the time and effort spent in reviewing our submission, and for the positive evaluation. Your comments make it clear that you understood and agreed with our motivation behind this project, for which we are humbled and extremely grateful. We are also very grateful for the suggestions for improvements, which we will take into account for future versions of this paper. We will do our best to address your concerns below.
>
> **Motivation**
>
> We fully agree that, as currently written, the axioms somewhat seem to be pulled out of the hat. In fact, these axioms were a result of over a year of contemplating different ways of axiomatising causality. Given more space, you are absolutely right that more discussion of the motivation and potential alternative axiomatisations would certainly benefit the paper, and we propose to do so in the future versions of this work.
>
> **Lines 137-138**
>
> The causal kernel $K_S$ is a transition probability kernel from $(\Omega,\mathscr{H}_S)$ into $(\Omega,\mathscr{H})$, so for a given $A\in\mathscr{H}$, the map $\omega\mapsto K_S(\omega,A)$ has to be measurable with respect to $\mathscr{H}_S$ (see definition of transition kernels in the appendix, lines 568-575). If $K_S$ depended on the $T\setminus S$ components, it would violate this measurability condition. We hope we understood your point correctly, and that this addresses your concern.
>
> **Minor comments**
>
> 1. We give the definition of measurable rectangles in the appendix on line 523. In future versions of this paper, we propose to move this definition to the main body in the camera-ready version.
>
> 2. You are right. We propose to change this, so that we make it clear that only the $\sigma$-algebra is in the form of a product, not the measure. Thank you very much for pointing this out - this is a great spot.
>
> 3. Actually (2) is not a product of kernels, because $K_{S\cup U}$ has the $\omega_{S\setminus U}$ component that does not get integrated with respect to the measure $L_{S\cap U}(\omega_{S\cap U},\cdot)$. In the appendix, Remark C.1(b) treats the special case in which the interventional kernel $K^{\text{do}(U,\mathbb{Q},\mathbb{L})}_S$ is a product of kernels.
>
> **Questions**
>
> I'm guessing you mean something like [Cinlar, 2011, page 151, Theorem IV.2.7]? If so, yes, we agree that this is definitely an interesting question to answer in future research. Thank you very much for the suggestion, we really appreciate it!

---

> > ### Comment · Reviewer_kUcy · 2023-08-18
> >
> > Thanks for the clarifications. I stand by my high rating of this paper.

---

### Official Review · Reviewer_pcib · 2023-07-02

**Soundness:** 4 excellent
**Presentation:** 3 good
**Contribution:** 4 excellent
**Rating:** 8
**Confidence:** 4

**Summary:**

This paper presents a completely novel mathematical description of causal systems, which generalizes the most well-known causal formalisms such as SCMs and potential outcomes. It does so by taking the probability distributions as the foundation, and use this to construct a measure-theoretic approach to causal systems. The result is a very expressive framework, that definitely has its place within the landscape of causal formalisms.


**Strengths:**

The paper is clearly written, but some background in probability theory is required to understand the technical definitions. There is substantial discussion of related work and open questions, illustrations of the framework, and comparisons to SCMs. Overall this is a great paper.


**Weaknesses:**

See questions.

Typos:

58: “we have designated” -> “we have a designated”

81: “implications concepts” -> “implications for concepts”

107: “the measurable space are” -> “the measurable space is”


**Questions:**

I do have some conceptual comments about the motivation and supposed superiority of the framework.

1: In your comparison with SCMs, you only consider hard interventions. What about soft interventions?

2: Remark 3.2:
I find this quite strange. Usually one thinks of an intervention not as an intervention on the system that affects a subsystem, but simply as an intervention on the subsystem itself, and this of course affects the system, just as changes to a part are also changes to a whole. I’m not saying the idea of a “holistic” intervention on an entire system is uninteresting, but merely that this is not usually how causal interventions are understood. In fact, the independent mechanism assumption made by Woodward and taken over by others captures a fundamental modularity that is assumed to hold for causal relations.

Similarly, the idea of coupling the observational and the interventional distributions is a feature, not a bug: it encodes an assumption about how we take the world to work.

In sum, although I agree that it is interesting to decouple these distributions and to view interventions as “system-wide”, as your formalism does, I am not sure when and why we would ever want to do so: at heart, the motivation for your framework seems to come with a philosophical theory that is lurking in the background which appears in conflict with the standard interventionist theory of Woodward that causal modellers usually align with. I realize that it would be too much for one paper to introduce both such a rich framework and dive into a philosophical analysis, but my suggestion would then be to here remain more neutral regarding these commitments.

3: footnote 1: do you think your framework is strictly more expressive than GSEMS? They don’t use probabilities, but perhaps you could reconstruct GSEMS by giving positive support to all values, and reasoning about P > 0?

4: Confounders.
I’m not sure if the authors are aware, but the philosopher Nancy Cartwright has famously argued against the Causal Markov Condition by using a very similar example, the so-called chemical factory. In her case though, she goes even further, stipulating that there is no common cause, not even an unobserved one. If one accepts her stipulation, then CBNs seem indeed unable to express such situations, whereas your framework would have no trouble with it at all. So it might be useful to refer to this discussion. For what it’s worth, the usual reply is similar to the move that you here discuss: simply add a “fictional” exogenous variable that functions as a latent common cause.

This brings me to footnote 2: I do not agree that such a variable is “completely meaningless”. It simply captures our ignorance. As long as one is committed to the Causal Markov Condition, one is committed to the belief that such a latent variable must exist. Whether we call it one variable or several variables collapsed into one seems merely a matter of terminology.

5: “Of course, cycling relationships abound in the real world.”
Really? That is quite a controversial claim: do the authors then subscribe to backwards causation? I’d say that cyclic models are useful abstractions of what in the end are non-cyclic systems. Your example is a case in point: it’s not the current price which causes the current amount of rice, and vice versa, it’s the price right now that determines the future amount.

Moreover, although I’m sure the authors are right that SCMs have trouble in representing certain cyclic cases, is this really one of them? It doesn’t seem so hard to model this with an SCM.


**Limitations:**

yes

---

> ### Author Rebuttal · Authors · 2023-08-05
>
> We thank the reviewer very much for the time and effort spent in reviewing our submission, and for the positive evaluation. Thank you also for suggestions for improvements; we will reflect them in future versions of this work. We will do our best below to address your concerns in the Questions section.
>
> 1. If we understand correctly, you are referring to Section 3.1, lines 214-215. This is not referring exclusively to hard interventions, it is simply constructing the causal kernels. Once these causal kernels are constructed, by placing a non-Dirac measure on the sub-$\sigma$-algebra on which we want to intervene, and also by utilising the internal causal mechanisms $L$, we can then formulate soft interventions. For future versions of this work, we propose to spell this out more explicitly.
>
> 2. Actually the point we were trying to make was precisely in agreement with your view - that we are most often interested in intervening on subsystems, rather than the whole system. Our causal kernels capture precisely this information, i.e. the effect of intervening on subsystems on the whole system. The point we were trying to make was that in the SCM formulation, the mathematics seems to go in the reverse direction to this philosophy, because each structural equation $f_j$ encodes information about the effect of intervening on the whole system on the subsystem $X_j$. Our causal space formalism does *not* view interventions system-wide - each causal kernel encodes information about intervening on the corresponding subsystem (sub-$\sigma$-algebra). In this sense, we are in complete agreement in terms of the philosophy (with Woodward, and that behind e.g. the SCM framework), we just differ in how to encode this mathematically.
>
> As for the decoupling of observational and interventional measures, we maintain that it is an advantage to be able to do this for the sake of generality. In order to be able to couple them, as it is done in SCMs, we have to impose the assumption that all of the variables have been included in the model that are required to do this (see Example 4.1). We do treat the case in which observational measure and the interventional measure (at least partially) match as an important special case - see Appendix D on the concept of sources that we introduce there.
>
> 3. Some philosophy behind causal spaces is shared with GSEMs in the sense that GSEMs also encode information about what happens to the whole system when intervening on a subsystem. However, there are some technical hurdles to be overcome if one wants to actually reconstruct a GSEM from a causal space. First of all, they consider power sets of the range rather than $\sigma$-algebras, which, if the variables are allowed to take uncountably many values, causes measure-theoretic problems. Moreover, as you rightly point out, for a given probability measure on the exogenous variables and an intervention, there seems to be no way of propagating this measure on the exogenous variables to sets of values of the endogenous variables.
>
> In the sense that GSEMs cannot do probabilities and causal spaces are designed explicitly with probabilities in mind, we believe that causal spaces strictly generalise GSEMs. We also do not believe that they worked with power sets rather than $\sigma$-algebras for any deep reason, and if we wrote out the GSEM formulation with $\sigma$-algebras, then indeed, as you suggest, we believe it would be possible to reconstruct GSEMs from causal spaces by letting the function $\mathbf{F}$ map to the support of each intervention. However, for reasons laid out above, there would be no way of uniquely reconstructing causal spaces from GSEMs.
>
> 4. This would be very valuable addition to our work, and we are very grateful to you for this comment. We absolutely propose to add a discussion on this example in future versions of this work.
>
> As for footnote 2, we agree with you that the variable itself is not meaningless. The variable of course can have meaning as a way of capturing our ignorance, precisely as you said. Our point was that such a variable cannot have numerical values and distributions that represent something meaningful. What does it mean for a variable that we name "everything in the world that we do not know about" to have value 1? What does it mean for it to be normally distributed? Properly defined random variables should be more concrete, e.g. height of a student in cm, temperature in Celsius, crop yield in tonnes, etc.
>
> 5. Thank you for raising this point. Please see author rebuttal to all reviewers.
>
> We thank you again for your positive review and suggestions, and should you have any more questions or if we misunderstood any of your points, we would be delighted to engage in the author-reviewer discussions.

---

> > ### Comment · Reviewer_pcib · 2023-08-11
> >
> > I have read the rebuttal. Thanks for the clarifications!

---

### Official Review · Reviewer_aMnb · 2023-07-07

**Soundness:** 4 excellent
**Presentation:** 3 good
**Contribution:** 3 good
**Rating:** 7
**Confidence:** 4

**Summary:**

The paper develops new foundations for causality based on the measure-theoretic foundations of probability pioneered by Kolmogorov. Interventions are formalized as transition kernels and it's shown that the new approach is more general than existing ones in a number of respects.

**Strengths:**

The paper is clearly laid out, sound, and represents a noteworthy (and as far as I know, novel) attempt to do causality in terms of mathematical probability theory. I believe the most promising direction in terms of utility is in being able to handle continuous-time causal stochastic processes, which have so far resisted formalization; the approach here does not even rely on dynamical systems.

**Weaknesses:**

I raise some edits and additional citations below.

**Questions:**

- Line 49: Why is there no citation to Ibeling & Icard's 2020 "Probabilistic Reasoning across the Causal Hierarchy"? This paper was (one of the, if not the) first to axiomatize SCMs in a truly *probabilistic* setting. The references [18, 20] there are still only quasi-probabilistic. (Admittedly, the paper I suggest does enforce an acyclicity constraint, but it is no better or worse than the other two references in this regard.)
- Line 296: Citation 20 probably should also be thrown in here.
- Can the rice example be modeled within the space of SCMs that always have unique solutions? I can't see why not. In this particular case, although the framework here may be more natural, it may not add much that is substantive.
- Example 4.2: notations K_1(3, x) and K_2(6, x) seem inconsistent with each other since the interventions are on different variables.
- Can the authors comment on whether the framework given invalidates quintessential principles of SCMs, e.g., effectiveness, composition, and reversibility? I am guessing the answers are no (by interventional determinism, line 158), yes, yes. The broader point is that not only do SCMs require additional assumptions to be well-defined, but that these assumptions may also reverberate in causal reasoning.
- Line 328: Z = {..., 2, 1, 0, 1, 2, ...} should read Z = {..., -2, -1, 0, 1, 2, ...}

**Limitations:**

Yes, they have.

---

> ### Author Rebuttal · Authors · 2023-08-05
>
> We thank the reviewer for the time and effort spent in reviewing our submission, and for the overall positive evaluation. We also thank you for raising interesting points and suggestions for improvements, which we will take into account in future versions of this work. Below, we will try our best to address your concerns and queries.
>
> **Line 49**
>
> This is a very reasonable citation recommendation. We had of course read the paper during the project, but inexplicably left out its citation in our submission. We will make sure to include it in future versions of this work. Thank you very much for pointing it out.
>
> **Line 296**
>
> Agreed - thank you very much!
>
> **Rice Example**
>
> Thank you for making this point - please see author rebuttal to all reviewers.
>
> **Example 4.2**
>
> Thank you for pointing this out. We will change one of the $x$'s into a $y$ for better notation.
>
> **Principles of SCMs**
>
> By effectiveness, composition and reversibility, we assume that the reviewer is referring to the notions introduced under these names in [Galles and Pearl, 1998, An Axiomatic Characterisation of Causal Counterfactuals, Section 3]. This is a very interesting point to consider, and we are very grateful to you for raising it.
>
> Firstly, even though the concepts of effectiveness, composition and reversibility can be carried over to causal spaces, the mathematics through which they are represented needs to be adapted, since the tools that are used in causal spaces are different from those used in causal models of [Galles and Pearl, 1998]. In particular, we work directly with *measures* as the primitive objects, whereas [Galles and Pearl, 1998] use the structural equations as the primitive objects, and the probabilities only enter through a measure on the exogenous variables. Thus, the three properties can be phrased in the causal space language as follows:
>
> Effectiveness: For $S\subseteq R\subseteq T$, if we intervene on $\mathscr{H}_R$ via a measure $\mathbb{Q}$, then $\mathscr{H}_S$ has measure $\mathbb{Q}$ restricted to $\mathscr{H}_S$. This is indeed guaranteed by interventional determinism (Definition 2.2(ii)), as you said.
>
> Composition: For $S,R\subseteq T$, denote by $\mathbb{Q}'$ the measure on $\mathscr{H}_{S\cup R}$ obtained by restricting $\mathbb{P}^{\text{do}(S,\mathbb{Q})}$. Then $\mathbb{P}^{\text{do}(S,\mathbb{Q})}=\mathbb{P}^{\text{do}(S\cup R,\mathbb{Q}')}$. In words, intervening on $\mathscr{H}_S$ via the measure $\mathbb{Q}$ is the same as intervening on $S\cup R$ via the measure that it would have if we intervened on $\mathscr{H}_S$ via $\mathbb{Q}$. This is not in general true, as you said. A counterexample can be demonstrated with a simple SCM, where $X_1$, $X_2$ and $X_3$ causally affect $Y$, in a way that depends not only on the marginal distributions of $X_1$, $X_2$ and $X_3$ but their joint distribution, and $X_1$, $X_2$ and $X_3$ have no causal relationships among them. Then intervening on $X_1$ with some measure $\mathbb{Q}$ cannot be the same as intervening on $X_1$ and $X_2$ with $\mathbb{Q}\otimes\mathbb{P}$, since such an intervention would change the joint distribution of $X_1$, $X_2$ and $X_3$ even if we give them the same marginal distributions.
>
> Reversibility: For $S,R,U\subseteq T$, let $\mathbb{Q}$ be some measure on $\mathscr{H}_S$, and $\mathbb{Q}_1$ and $\mathbb{Q}_2$ be measures on $S\cup R$ and $S\cup U$ respectively such that they coincide with $\mathbb{Q}$ when restricted to $\mathscr{H}_S$. Then if $\mathbb{P}^{\text{do}(S\cup R,\mathbb{Q}_1)}(B)=\mathbb{Q}_2(B)$ for all $B\in\mathscr{H}_U$ and if $\mathbb{P}^{\text{do}(S\cup U,\mathbb{Q}_2)}(C)=\mathbb{Q}_1(C)$ for all $C\in\mathscr{H}_R$, then $\mathbb{P}^{\text{do}(S,\mathbb{Q})}(A)=\mathbb{Q}_1(A)$ for all $A\in\mathscr{H}_R$. We again agree with you that this does not hold in general in causal spaces. In fact, Example 4.2 in our paper is a counterexample of this, with $S=\emptyset$.
>
> So all in all, we agree with your answers of no, yes, yes, as to whether the causal space framework challenges these fundamental properties of SCMs. Composition is an interesting one, because even in the SCM setting, it is very clear that it does not hold if we are interested in the *measure* of the target variable, not just the pointwise evaluation of the structural equations. This is a very interesting discussion, and we propose to include it in future versions of this work. Thank you very much again for raising this point!
>
> **Line 328**
>
> This is a terrible typo. Thank you very much for pointing it out.
>
> We thank you again for the review, positive evaluation and valuable suggestions for improvements. We hope to have clarified your questions, and should you have any further questions, or if we misunderstood any of your points, we would be happy to engage in the author-reviewer discussion.

---

> > ### Comment · Reviewer_aMnb · 2023-08-14
> >
> > Thanks, the axiomatic discussion is interesting and of course I agree it should be included. To the extent that a completeness proof could be given it might shed light on the question about the rice example (namely, if these principles are complete and the rice distribution obeys them, then there must exist a cyclic SCM modeling the distribution even if it seems difficult to model). I maintain my overall positive rating.

---

### Official Review · Reviewer_xRcV · 2023-07-11

**Soundness:** 2 fair
**Presentation:** 2 fair
**Contribution:** 2 fair
**Rating:** 4
**Confidence:** 3

**Summary:**

This paper proposes a measure-theoretical axiomatization of causalality with the notion of causal space and with a collection of causal kernels encoding the causal information.

**Strengths:**

The paper offers an axiomatization of causality which is based on the measure-theoretical foundation of probabaility theory.

**Weaknesses:**

(1) It seems that Definiiton 2.2 is not well-defined because, given a subset S of T, we cannot decide the causal mechanism K_S. Also see my questions below.
(2)  I really could not understand the claim (Line 147) that "intervention is the process of placing any desired measure,... along with an internal causal mechanism ...".   Then it seems that intervention can also be treated as conditioning.

**Questions:**

(1) I am confused by Definition 2.2 (Causal space).    For a given S, the causal mechanism seems undecided: there are many causal mechanisms satisfying the two axioms.  For two subsets S1 and S2 of the T, What is the relationship among K_{S1}, K_{S2} and K_{S1\cup S2}?
(2) It seems that all varaibles are independent in Definition 2.2.   How can one formulate the relationship that variable X causes another variable Y in the causal spapce?

**Limitations:**

yes

---

> ### Author Rebuttal · Authors · 2023-08-04
>
> Thank you for reviewing our submission. We regret that you have a rather negative view of the paper, and we hope that our clarifications below, as well as the other reviews, can give you a more positive view.
>
> **Definition 2.2**
>
> This is the main definition of the paper, containing the two axioms of causal kernels. We regret that you were not convinced by these axioms of causal spaces. We'll do our best to address your concerns.
>
> For a given $S$, of course there are many possible causal kernels $K_S$ that satisfy the two axioms, but in the end the causal space will have one $K_S$ specified. This is the same in SCMs, where, given a random variable (or a node), there are many possible functions that can act as the structural equation for this variable, but in the end the SCM will specify one function. Or even just in probability spaces, where of course there are many possible probability measures that satisfy the axioms of a probability measure, but a given probability space will have one measure specified.
>
> For two subsets $S_1,S_2$ of $T$, there is a priori no relationship between $K_{S_1}$, $K_{S_2}$ and $K_{S_1\cup S_2}$, and for full generality, this is desirable, since, even in the SCM framework, we can easily construct examples where specifying $K_{S_1}$ and $K_{S_2}$ does not fully determine $K_{S_1\cup S_2}$.
>
> We are not sure what you mean by the variables being independent in causal spaces, but it is easy to formulate a two-variable situation in the causal space framework with $X$ causing $Y$. This is precisely what was illustrated in Example 2.5.
>
> **Line 147**
>
> The fact that we take this view on intervention is not a "claim", but the philosophy behind our definition of intervention, and it agrees with most other definitions of intervention in the literature, including the SCM case. We pick a subset of the variables on which we would like to intervene (most often a single variable), and we place a desired measure on those (that) variable(s) (for hard interventions, a Dirac measure). The causal components in the frameworks (the structural equations in SCMs, the causal kernels in our causal spaces) then determine what happens to what we did not intervene on.
>
> In our causal space framework (and in any other framework), intervention is categorically not the same as conditioning - in fact this is the whole point of having any theory of causality. In examples throughout our submission, there are many instances in which intervention is not the same as conditioning, showing that our causal space framework accommodates this. For example, in Example 2.5, conditioning on temperature is not the same as intervening on temperature, and in Example 4.4 and Figure 5, conditioning on the Brownian motion to have a particular value at a particular time point is not the same as intervening on the Brownian motion to have a particular value at a particular time point.
>
> We thank you again for your review, and we hope that our rebuttal, along with the opinions of the other reviewers, are sufficient to convince you at least of the validity of causal spaces, and if possible, their merits. Please let us know if you have further questions, or if any of our clarifications were unclear.

---

> > ### Comment · Area_Chair_KtBo · 2023-08-14
> >
> > Dear reviewer xRcV,
> >
> > You have been the most negative reviewer for this paper. Can you please respond to the authors rebuttal, and also take the other reviews into account?
> >
> > Thank you, AC

---

### Official Review · Reviewer_TfjY · 2023-07-16

**Soundness:** 4 excellent
**Presentation:** 3 good
**Contribution:** 3 good
**Rating:** 7
**Confidence:** 3

**Summary:**

(Preamble warning: given the amount and time I had to review Neurips papers, and given other duties, I focused solely on the main content of the paper, not consulting the appendices that were actually longer than the paper.)

The paper proposes and discusses a new framework, based on measure-theoretic probabilities, to encode causal systems, both on the observational and interventional level.

This is done through the use of probability kernels and probability measures defined over Cartesian products of spaces. The framework generalizes a number previously proposed frameworks,

**Strengths:**

* A unifying framework relying on measure-theoretic probabilities that is well-justified from a mathematical point of view, and can be summed up by two technical constraints on the system.

* A quite well-written paper, with illustrative examples showing how the framework can be applied.

**Weaknesses:**

* The authors make bold claims on probability theory: while it is quite okay to rely on measure-theoretic probabilities to derive a framework, I would argue that claiming that this view of probability is "near-undisputed and universally accepted" (P1, L24-25) and that "In ... probability theory, one starts by assuming the existence of a probability space" (P3, L101-102) is not true in the modern world. For instance and only to mention a few works, De Finetti betting accounts of probabilities allow for finitely additive probabilities, and does start by defining a probability space on a $\sigma$-algebra. Similarly, Shafer/Vovk account of probabilities from a game-theoretic point of view does not start from a measure-theoretic view point. Finally, one could also argue that causality could be seen through the lense of other uncertainty theories, and that the unversality of probability theory could be challenged itself (see works of Peter Walley on imprecise probabilities extending De Finetti's betting argument, and works of followers). So even if I think that the adopted viewpoint here is reasonable, I would avoid suggesting that it is the only one that can be considered, or that it can hardly be questioned.

* Sufficiency of the proposed axioms: the axioms are given as mathematical constraints over the system, and makes sense in a fully probabilistic setting (it is less clear if ones consider more expressive languages than probabilities). However, such kinds of axioms (unchange with respect to no new information + conformity with imposed constraints on the system) can also be found in other settings such as, e.g., Jeffrey's updating rule in the presence of uncertain information. I somehow miss a discussion of what makes those axioms peculiar to the situation of causalilty? Or does it follows from the assumptions that the probabilistic kernels DO model an exisiting cause rather than something else? And in this later case, one could imagine a situation where specified kernels do not encode actual causality relationships, while still following the proposed axioms. My feeling is that the axioms concern more mathematical properties ensuring that the system will behave well, rather than axioms ensuring that the encoded concepts and systems will indeed be causal in nature.

* Not clear whether more generality gives a better means to identify causation: In Example 4.1. I appreciate the fact that the proposed model is much more flexible than SCM, however this example raises the question of knowing whether the system is not to general to be able to identify causalities? More precisely, assume we observe the S-I correlation, as well as measure the temperature. If we focus on the SI, we then have two models that perfectly explains the data: one with no causation and observed correlation, and one with causation (where one would have to specify the causation mechanisms). In contrast, SCM would not be able to account for S-I correlation without assuming causaition, and would incitate the modeller to add potential explaining causations. So it is unclear whether the provided generality comes with better capacities of identifying potential causations in practical modelling situations?


**Questions:**

Questions:

*  P6, 240-243: what is meant by unique existence? Is it meant that a given observational/interventional system can only be represented by a unique set of measures and kernels? The exact meaning of uniqueness is unclear to me here.

* In example 4.2., I agree that one variable can influence the other (in a causal way), but I would argue that we are also faced here with a dynamical process (rice production can only be impacted by price for the next period of rice harvesting, and vice-versa). Would it be possible to provide an example of cyclicity where we are not faced with time-evolving processes?

* P8, L318: could you give some references to identify "by some authors"?

* Comutability: one question that arises from the presented framework (but is probably to be adressed in future work) is about the computability of the presented framework? Maybe it would be good to mention a few cases where this is achievable, if authors have already identified such cases.

* P9, L359-360: it is not entirely clear to me how the proposed axiomatization and the measure-theoretic view of probabilities are linked in the proposed framework. To be more precise, my feeling is that the two proposed axioms of no action-no causation and of agreement witht he causal mechanisms are not especially linked to a measure-theoretic view, which offers the technical tools to derive it. For example: to which extent do we really need $\sigma$-additive measures rather than finitely additive ones to express these two axioms? Of course the first option is technically/pragmatically more simple to deal with, but I am not completely convinced that this is at the heart of the expressed axioms.

Suggestions:

* Figure 1: suggesting that probability theory is confined to modelling data-generating process is a bit misleading, as a Bayesian setting allows one to put probabilities on any quantity of interested (including those not generated by data).

* Given this absence of discussions about the philosohical concepts of causality and how these axioms encode them, speaking of "Axiomatization of causality" seems a bit strong to me. Why not referring in the title to "Generalised causal model through measure-theory"?

* P3, L108-113: the vocabulary used here as well as the various differentiation made is a bit ambiguous: why speaking of "chains of trials" that make one think of repeated experiments when in all further examples all measurable spaces belong to distinct variables?

* P5, L195-196: an underlying assumption here is that the causal kernels are 1. perfectly specified and 2. encode causal relationship. In practice, nothing prevents those two points to be false, and I think it should be mentioned somewhere that those assumptions are made.



**Limitations:**

Limitations were adequately adressed.

---

> ### Author Rebuttal · Authors · 2023-08-04
>
> Thank you for your time and effort in reviewing our submission; we are very grateful for your overall positive evaluation of the paper and your valuable suggestions. We will do our best below to address your concerns and queries.
>
> **Weaknesses**
>
> Thank you for this point. We are aware that other theories of probability exist other than Kolmogorov's. Of those you mentioned, we are aware of de Finetti's approach, and are vaguely aware of Shafer/Vovk's game theoretic framework. We were not aware of Walley's work - thank you for pointing it out. There is also an approach more amenable to Bayesian probability ([Jaynes and Bretthorst, 2003], or https://en.wikipedia.org/wiki/Cox%27s_theorem). We acknowledge that these works are valuable, and we propose to make more of an effort to acknowledge them. We also propose to water down our language from "near-undisputed and universally accepted". However, we believe it is equally unreasonable to treat Kolmogorov's framework as simply one out of many on the same level of significance, given that the vast majority of research in probability theory, as well as the vast majority of textbooks and lecture courses on probability theory, is built on it.
>
> We agree that the two axioms we provide are a priori mathematical properties that are not in themselves causal in nature. However, this is also true for probability spaces - they are simply measure spaces where the whole set happens to have measure 1. Same for densities - they are simply functions that happen to integrate to 1. Yet, we give these objects a probabilistic interpretation. Our axioms of causal kernels should be viewed in the same light.
>
> Thank you for raising this point. We agree that in general identification is difficult, if not impossible, with such generality. However, we would like to point out that if measuring temperature is possible, and if it is in our interest to do so (for identification or any other reason), then our causal space framework accommodates that, so in this sense, nothing is "lost" compared to the SCM framework. We maintain that this is advantageous in the modelling perspective over SCMs, where the researcher is "forced" to include other variables that explain the correlation. For identification from data in our framework, Section D in the Appendix is dedicated to it, via a concept which we call "sources".
>
> **Questions**
>
> P6, 240-243: Uniqueness here is meant simply to contrast with SCMs, where, without the acyclicity assumption, there may be many distributions over the endogeneous variables that satisfy the structural equations and the noise variables (see, for example, [Bongers et al., 2021, Foundations of Structural Causal Models with Cycles and Latent Variables]). In causal spaces, the observational and interventional distributions are always uniquely defined.
>
> Example 4.2: Thank you for raising this point. Please see author rebuttal to all authors.
>
> P8, 318: For example, [Peters et al., 2017, Elements of Causal Inference, Remark 6.5]. We agree that we should give some citation here, and we propose to do so in future versions of this work.
>
> Computability: This is not a question that we had thought to address in this submission at all, but we agree that this is a crucial and interesting research direction. We are grateful to you for pointing it out.
>
> P9, 359-360: We agree that the ideas behind the two axioms we give are not inherently measure-theoretic, and that it would probably be possible to endow other frameworks (e.g. graphical models) with these concepts. However, as you point out, the way these axioms represent these concepts are measure-theoretic, since the transition kernels are very much measure-theoretic tools.
>
> **Suggestions**
>
> Figure 1: In terms of what is included in the outcome space, the parameters in Bayesian setting could also be viewed as "data". But we see the point you are trying to raise. What would be your suggestion for the left-hand box?
>
> The focus of this work was not the philosophical discussions around causality on the level of a philosophy paper, which we hope you understand. However, we maintain that calling our proposal an "axiomatisation" is justified in the mathematical sense, in that we lay out what we assume to be true about causal spaces, and we build subsequent definitions and results on these axioms. This is precisely in the form of other "axiomatic frameworks" in mathematics, not least the probability axioms of Kolmogorov. Moreover, throughout the text we clearly state what philosophical approach behind causality we take, namely that we view it as a study of what happens when we intervene, and in Remark 2.4, we discuss precisely what our axioms encode, so we do not fully agree with your judgment that there is an "absence of discussions about the philosohical concepts of causality and how these axioms encode them".
>
> P3, L108-113: Thank you for this suggestion, yes, we agree that "chain of trials" is somewhat misleading. The terminology is taken from [Cinlar, 2011, Probability and Stochastics, page 161]. Of course, the mathematics $(E_t,\mathcal{E}_t)$ allows for distinct variables even here. We propose to change it to "set of trials", would you agree that this is better?
>
> P5, L195-196: We would again like to draw parallels with probability spaces, where, of course, it is equally impossible to perfectly specify the probability measure in practice for any given problem, but we still do so in the modelling process. In the same vein, we feel it is justified to assume that for the causal kernels. We will make this point more explicit.
>
> We thank you again for your positive review, and raising many interesting points. If accepted, your suggestions are sure to improve our paper. If any of our answers were unclear or we misunderstood your point, please let us know. We would be happy to engage in further discussions.

---

> > ### Comment · Reviewer_TfjY · 2023-08-17
> > **Thank you for the discussion**
> >
> > Dear Authors,
> >
> > Thanks for the nice answers and discussion.
> >
> > About the little details:
> >
> > * P3, L108-113: I would say that the name trial evoke repeated experiments in my mind, and I am not sure it quite fits the proposed framework. However, if this is the vocabulary used in some previous similar settings, I have no strong arguments against it.
> >
> > * Figure 1: I would say that if one has graphical models and subjective trends of probabilities in mind, one could probably replace data by evidence and data-generating process by knowledge model or something similar (population model?). However, I also have no strong oppinion aobut that, it is just that confining probability and causality to data and data-generation process (which suggest a frequentist flavour) seems a bit limiting (yet sufficient anyway to encompass most if not all ML problems).
> >
> > * About the measure-theoretic approach and the axiomatic: thank you for the discussion here, and I somehow agree with the points made about the use of axiomatisation and the fact that Kolmogorov measure-theoretic approach is the most used and wide-spred. However, an approach/intrepretation being the main one is not really an argument to support the fact that it is the most relevant one to treat/view a given problem. And I would argue that, in the case of causality, having a clear semantic/interpretation of the probabilities and probabilistic model used seems important, hence my hope for such a discussion. I am quite fine with purely formal/mathematical characterisation theorems, and I do appreciate the work presented here. I guess more philosophical discussions can be postponed to future times.

---

> > > ### Author Response · Authors · 2023-08-18
> > > **Thank you for your response**
> > >
> > > Dear reviewer TfjY,
> > >
> > > Thank you for your thoughtful response to our rebuttal.
> > >
> > > We agree that the word trial does make one think of repeated experiments. We'll try to think of some other wording that is more suitable. Perhaps occurrences?
> > >
> > > We also agree with the second point, and we think "population model" is a great suggestion, and certainly sounds more fitting than the data generating process, especially if we do not restrict ourselves to the frequentist mindset. We'll make this change in the future versions of this work.
> > >
> > > Thank you for providing further discussion on this important point. We agree that, if one wants to extend a theory of probabilities to a theory of causality, then one must carefully consider the merits of each framework of probability for that goal, and Kolmogorov's framework shouldn't be the automatic choice purely on the basis of its widespread acceptance. We must admit that we have not given this as much deliberation as we perhaps should have done, and as written in the original rebuttal, we propose to give more of a discussion on this point. Thank you also for reiterating your appreciation of our work, we don't take it for granted and we are hugely grateful.

---

### Author Rebuttal · Authors · 2023-08-06

We would like to thank all reviewers for their time and effort in reviewing our paper, and for making many valuable suggestions that are sure to improve the draft. We are very grateful for your time, and do not take it for granted. We are also very grateful for the mostly positive reviews, and kind words. We answer points raised by individual reviewers separately, except those relating to cycles and Example 4.2, which we answer jointly here.

**Do cyclic causal relationships truly exist, or is everything acyclic if we include a time component?**

As challenged by reviewer TfjY, it is difficult to think of truly instantaneous cyclic causal relationships (unless they are random variables that do not live in separate components of a product $\sigma$-algebra; see Remark 2.7(i). For example, if $X$ and $Y$ represent the weight of a box in kg and stones respectively, then one could argue that $X$ "causally" affects $Y$ and vice versa, and that this causal effect is instantaneous. However, in our opinion, it makes no sense to talk about the causal effect between random variables that do not live in separate components of a product $\sigma$-algebra. In SCMs, or in any other causal model based on graphs, random variables represented by each node should live in separate components of a product $\sigma$-algebra. Note that this is different to when two random variables do live in separate components of a product $\sigma$-algebra but a measure *happens* to be supported on some kind of a diagonal).

We believe that even in non-cyclic situations, most causal relationships have a hidden time component, although one could argue that some causal effects really are instantaneous, for example, altitude causally affecting temperature, or the picture of a cow causally affecting its label to be "cow". Our stance is more aligned with reviewer pcib, in that modelling cyclic causal relationships are useful abstractions of what are in the end non-cyclic causal relationships. We also believe that it is extremely difficult to argue against the necessity of such an abstraction, if researchers are happy to model non-cyclic, time-dependent causal relationships without explicitly modelling the time component (as it is in fact very often done) and if we believe in the necessity of being able to do so. It is in this sense that we wrote "cyclic causal relationships abound in the real world", although, as pointed out by reviewer pcib, this sentence may be somewhat misleading. We propose to replace it with something that better reflects the discussion in this paragraph. We also do not want to completely rule out the possibility that one might come across a truly instantaneous and cyclic causal relationship, although we are leaning more towards the belief that such a situation may not exist.

On the other hand, if one were to insist on modelling the time component explicitly every time it is present (which is not our stance), then we argue that the current tools to model causality within stochastic processes fall short, and we believe that causal spaces make significant contribution in this regard too, as argued in Section 4.3 and as agreed by reviewer aMnb. In answer to reviewer pcib, no, we do not subscribe to causation that goes backwards in time, and it is for this reason that we felt it important to introduce the concept of *time-respecting causal mechanism* in Definition 4.3.

**Example 4.2**

Reviewers aMnb and pcib suggested that it would not be difficult to model the situation in Example 4.2 with a solvable cyclic SCM [Bongers et al., 2021]. To us, this was difficult, and we didn't manage to do so, both when we were writing the paper and during this rebuttal period. One of the reasons is that the observational measure and the interventional measures are decoupled, and the interventional measure (after intervening on either variable) does not equal the corresponding conditional measure, which means that we must include a common hidden confounding variable. But even apart from this, it seems like a difficult task (at least for us; perhaps we lack the skills or experience for it) to set up the noise variables and their distributions, and to come up with structural equations that yield given desired observational and interventional measures. The example given in [Bongers et al., 2021, Example 3.5] seems like an extremely simple case. We admit that it could still be possible and that we just lack the skills to do it; equally, we did not show that this was *not* possible, and we would be curious to know if the reviewers were able to do this.

But it should at least serve to illustrate one point - that in SCMs, if we start with noise distributions and structural equations that involve cycles, then it is highly non-trivial to show the existence and uniqueness of a solution, and if we start with the observational and interventional distributions of the endogenous variables, then it is (at least for us) very difficult to find the noise distributions and the structural equations that yield them as a (unique) solution. On the other hand, we argue that causal spaces facilitate a much more natural expression of cyclic causal relationships.

We would like to thank all reviewers again for their time, and if you have any further questions, or if any of our explanations were unclear, or if we misunderstood any of your points, we would be very grateful if you could let us know. We look forward to engaging in further discussions in the author-reviewer discussion period.

---

### Decision · Program_Chairs · 2023-09-21

**Decision:**

Accept (oral)

**Comment:**

**This is a very strong paper on the measure theoretic foundation of causality and highly important contribution to the field. Deserves special attention, such as an oral and/or an award (in the theory segment, if there are different categories).**

There was one negative reviewer (xRcV), but the review was short and rather uninformative. The reviewer also disappeared and did not react, neither to requests from the authors nor the AC. The review should therefore be disregarded.